EMBO
Molecular Medicine

# Hepato-entrained B220[+]CD11c[+]NK1.1[+] cells regulate pre-metastatic niche formation in the lung

Sachie Hiratsuka[1,2,*,†] (iD), Takeshi Tomita[1], Taishi Mishima[1], Yuta Matsunaga[1], Tsutomu Omori[1], Sachie Ishibashi[1], Satoshi Yamaguchi[3], Tsuyoshi Hosogane[3], Hiroshi Watarai[4], Miyuki Omori-Miyake[5], Tomoko Yamamoto[6], Noriyuki Shibata[6], Akira Watanabe[7,‡], Hiroyuki Aburatani[7], Michio Tomura[8], Katherine A High[9] & Yoshiro Maru[1,**] (iD)

## Abstract

Primary tumours establish metastases by interfering with distinct organs. In pre-metastatic organs, a tumour-friendly microenvironment supports metastatic cells and is prepared by many factors including tissue resident cells, bone marrow-derived cells and abundant fibrinogen depositions. However, other components are unclear. Here, we show that a third organ, originally regarded as a bystander, plays an important role in metastasis by directly affecting the pre-metastatic soil. In our model system, the liver participated in lung metastasis as a leucocyte supplier. These liver-derived leucocytes displayed liver-like characteristics and, thus, were designated hepato-entrained leucocytes (HepELs). HepELs had high expression levels of coagulation factor X (FX) and vitronectin (Vtn) and relocated to fibrinogen-rich hyperpermeable regions in pre-metastatic lungs; the cells then switched their expression from Vtn to thrombospondin, both of which were fibrinogen-binding proteins. Cell surface marker analysis revealed that HepELs contained B220[+]CD11c[+]NK1.1[+] cells. In addition, an injection of B220[+]CD11c[+]NK1.1[+] cells successfully eliminated fibrinogen depositions in pre-metastatic lungs via FX. Moreover, B220[+]CD11c[+]NK1.1[+] cells demonstrated anti-metastatic tumour ability with IFNγ induction. These findings indicate that liver-primed B220[+]CD11c[+]NK1.1[+] cells suppress lung metastasis.

Keywords anti-metastasis; B220[+]CD11c[+]NK1.1[+] cells; coagulation factor; liver education; pre-metastatic lungs

Subject Categories Cancer; Digestive System; Immunology

## Introduction

Cancer remains a leading cause of deaths globally, with the number expected to increase to 21 million by 2030 (Vinay *et al*, 2015). Tumour metastases account for approximately 90% of all cancer-related deaths, although recent medical advances have aided many patients (Spano *et al*, 2012; Vinay *et al*, 2015).

Metastatic mobilization is generated by the triangular interplay among a primary tumour, a metastatic tissue and the bone marrow (Wels *et al*, 2008). Although the tumour-induced systemic environment likely affects metastases, the precise role of non-metastatic organs has yet to be elucidated. In some organs, the local microenvironment can be altered by a distant primary tumour resulting in conditions referred to as pre-metastatic soil, or niche (Hiratsuka *et al*, 2002, 2006, 2008; Kaplan *et al*, 2005; Erler *et al*, 2009; Huang *et al*, 2009; Jung *et al*, 2009; Kim *et al*, 2009; Hood *et al*, 2011; Peinado *et al*, 2012, 2017; Sceneay *et al*, 2013; McAllister & Weinberg, 2014; Costa-Silva *et al*, 2015). We have reported that the pre-metastatic niche in the lung occurs within distinct regions composed of fibrinogen deposition, resident tissue cells and infiltrating immune cells (Hiratsuka *et al*, 2013). Pre-metastatic organs

---

1 Department of Pharmacology, Tokyo Women's Medical University School of Medicine, Shinjuku-ku, Tokyo, Japan
2 PRESTO, Japan Science and Technology Agency (JST), Kawaguchi, Japan
3 Research Center for Advanced Science and Technology, The University of Tokyo, Tokyo, Japan
4 Division of Stem Cell Cellomics, The Institute of Medical Science of the University of Tokyo, Tokyo, Japan
5 Department of Microbiology and Immunology, Tokyo Women's Medical University School of Medicine, Tokyo, Japan
6 Department of Pathology, Tokyo Women's Medical University School of Medicine, Tokyo, Japan
7 Genome Science Division, Research Center for Advanced Science and Technology, The University of Tokyo, Tokyo, Japan
8 Laboratory of Immunology, Faculty of Pharmacy, Osaka Ohtani University, Osaka, Japan
9 Center for Cellular and Molecular Therapeutics, The Children's Hospital of Philadelphia, Philadelphia, PA, USA
*Corresponding author. Tel: +81 3 5269 7417; E-mail: nakamura.sachie@twmu.ac.jp
**Corresponding author. Tel: +81 3 5269 7417; E-mail: maru.yoshiro@twmu.ac.jp
†Present address: Department of Biochemistry and Molecular Biology, Shinshu University School of Medicine, Nagano, Japan
‡Present address: Genome/Epigenome Analysis Core Facility, Center for iPS Cell Research and Application, Kyoto University, Kyoto, Japan

---

contain a variety of resident tissue cells, as well as infiltrating bone marrow-derived cells (BMDCs). Importantly, the pre-metastatic soil contains a complex mixture of factors that either inhibit or promote metastasis (Qian & Pollard, 2010; Granot *et al*, 2011). In addition, post-metastatic regions contain more inflammatory immune cells in which pro-metastatic cells were activated rather than anti-metastatic cells (McAllister & Weinberg, 2014). Taken together, a distinct fibrinogen area may be a scaffold that attracts various immune cells as well as circulating tumour cells. However, it is not known whether any cell population can eliminate metastatic niche or combat cancer cells. Thus, we tried to find out anti-metastatic cells in pre- and post-metastatic niche.

In this paper, we found a specific immune cell population that was educated in liver, accumulated in lung niche via circulation from liver and capable of eliminating pre-metastatic fibrinogen deposition and killing metastatic tumour cells. In addition, application of those liver-primed cells efficiently functions with anti-metastatic ability.

## Results

### FX⁺CD45⁺ cells accumulate in pre-metastatic lung niche

To search for potential anti-metastatic factors, we sought functional cell populations related to fibrinogen clearance because the fibrinogen deposition area is scaffold for immune cells in pre-metastatic lungs (Hiratsuka *et al*, 2013). Fibrinogen is primarily produced in the liver and associated with a coagulation cascade; thus, we screened molecules related to liver-specific and coagulation system genes in circulating leucocytes of tumour-bearing mice. We generated tumour-bearing mice using E0771 breast cancer, LLC lung carcinoma and B16 melanoma cells. We used 6- to 8-week male and female mice for LLC and B16 and only female for E0771 tumours. In this study, a key point of our pre-metastatic model system is that spontaneous metastasis from the primary site was observed only after the primary tumour resection, although an intravenous injection of these cells easily attained lung metastasis (Hiratsuka *et al*, 2002, 2006, 2008) (see Materials and Methods). In addition, it should be noted that these tumour cells failed to metastasize to the liver. As shown in Appendix Table S1, tumour-bearing mice exhibited a threefold increase in coagulation factor X (*FX*, *F10*) expression in leucocytes compared with tumour-free controls. This trend was also apparent in CD45⁺ leucocytes derived from E0771 and LLC

tumour-bearing mice (Fig 1A). We next demonstrated that livers derived from tumour-bearing mice more significantly induced *FX* expression in CD45⁺ leucocytes than those from no tumour-bearing mice (Fig 1B). Immunostaining results confirmed the results also at protein levels (Fig 1C). In addition, bone marrow transplantation (BMT) strategy using GFP⁺-BM revealed that CD45⁺ leucocytes, which were derived from BM showed strong FX expression in tumour-bearing mouse liver (tumour-bearing mouse liver meaning liver derived from tumour-bearing mice) (Appendix Fig S1). Primary tumours induced lung fibrinogen depositions (Fig 1D, Appendix Fig S2), and accumulation of FX⁺CD45⁺ cells was detected in fibrinogen deposition areas in pre-metastatic lungs (Fig 1E). These results led us to hypothesize that the pre-metastatic liver induces FX⁺ leucocytes in tumour-bearing mice.

### FX⁺CD45⁺ cells contain B220⁺CD11c⁺NK1.1⁺ cells that relocate from liver to lung in tumour-bearing mouse

To examine cellular trafficking between the liver and pre-metastatic lungs of tumour-bearing mice, we developed an *in vivo* cell-tracking system using KikGR mice (Tomura *et al*, 2014). KikGR is a marker protein with a violet light-induced green-to-red photoconvertible fluorescent group. We prepared tumour-bearing KikGR mice using E0771 and LLC tumours. In addition, we used a tumour-conditioned media (TCM) injection technique to generate mice in the pre-metastatic phase, with no possibility of micrometastasis. For this model, we first applied violet light to a liver lobe of tumour-bearing- or TCM-stimulated KikGR mice (see model in Fig 2A, with further details in Materials and Methods). This irradiation protocol was established so that there would be no evidence of inflammation as monitored by the accumulation of CD11b⁺ cells and inflammation-related gene expression in violet light-treated mice (see details in Materials and Methods). Briefly, a small frontal area of the liver (circle area in 10 mm diameter, 100 μm depth) was exposed to violet light twice for 2 min while supplying phosphate-buffered saline (PBS). We kept this protocol because when we expanded area and time for irradiation to obtain more photoconverted cells, CD11b⁺ cell mobilization and inflammation-related gene expression were markedly stimulated. Blood perfusion was carried out before tissue collection to reduce the effect of circulating blood cells. The signal of KikGR red protein was clearly observed in the lungs 72 h after photoconversion, but not in non-photoconverted lungs (Fig 2B). The photoconverted cells were also detected by flow cytometric analysis (Fig EV1A, B and D, and Appendix Figs S3 and S4).

---

**Figure 1. Appearance of coagulation factor X (FX) positive-hepato-entrained leucocytes (HepELs) in peripheral blood and lungs during the pre-metastatic phase.**

A   Relative mRNA levels of *FX* in CD45⁺ leucocytes in the peripheral blood of E0771 (abbreviated to E) or LLC tumour-bearing mice. The mean sizes of E0771 and LLC tumours were 9.8 mm and 9.5 mm, respectively. Shown are averages (*N* = 8 mice/group) with SEM and one-way ANOVA.

B   Relative mRNA levels of *FX* in CD45⁺ leucocytes in various organs such as lung (Lu), liver (Li), spleen (Sp), bone marrow (BM) and lymph nodes (Lymph; inguinal (Ing) and mesenteric (Mes)) derived from no tumour-bearing or E0771-bearing mice. "Tumour" stands for mRNA levels of *FX* in E0771 tumours. Shown are averages (*N* = 6: organs from no tumour-bearing mice, *N* = 6: organs from E0771-bearing mice) with SEM and one-way ANOVA.

C   Immunohistochemical quantifications of FX expression in CD45⁺ leucocytes in the liver and lungs of no tumour-bearing or E0771-bearing mice. Shown are averages (*N* = 6) with SEM and Welch's *t*-test.

D   Immunohistochemical quantifications of fibrinogen deposition in no tumour-bearing and tumour-bearing mouse lungs. Shown are averages (*N* = 7/group, 14 random fields) with SEM. Welch's *t*-test.

E   Representative photographs of immunohistochemical co-localization of FX⁺CD45⁺ cells and fibrinogen deposition in tumour-bearing mouse lungs. Circles show high fibrinogen deposition areas (scale bar, 50 μm).

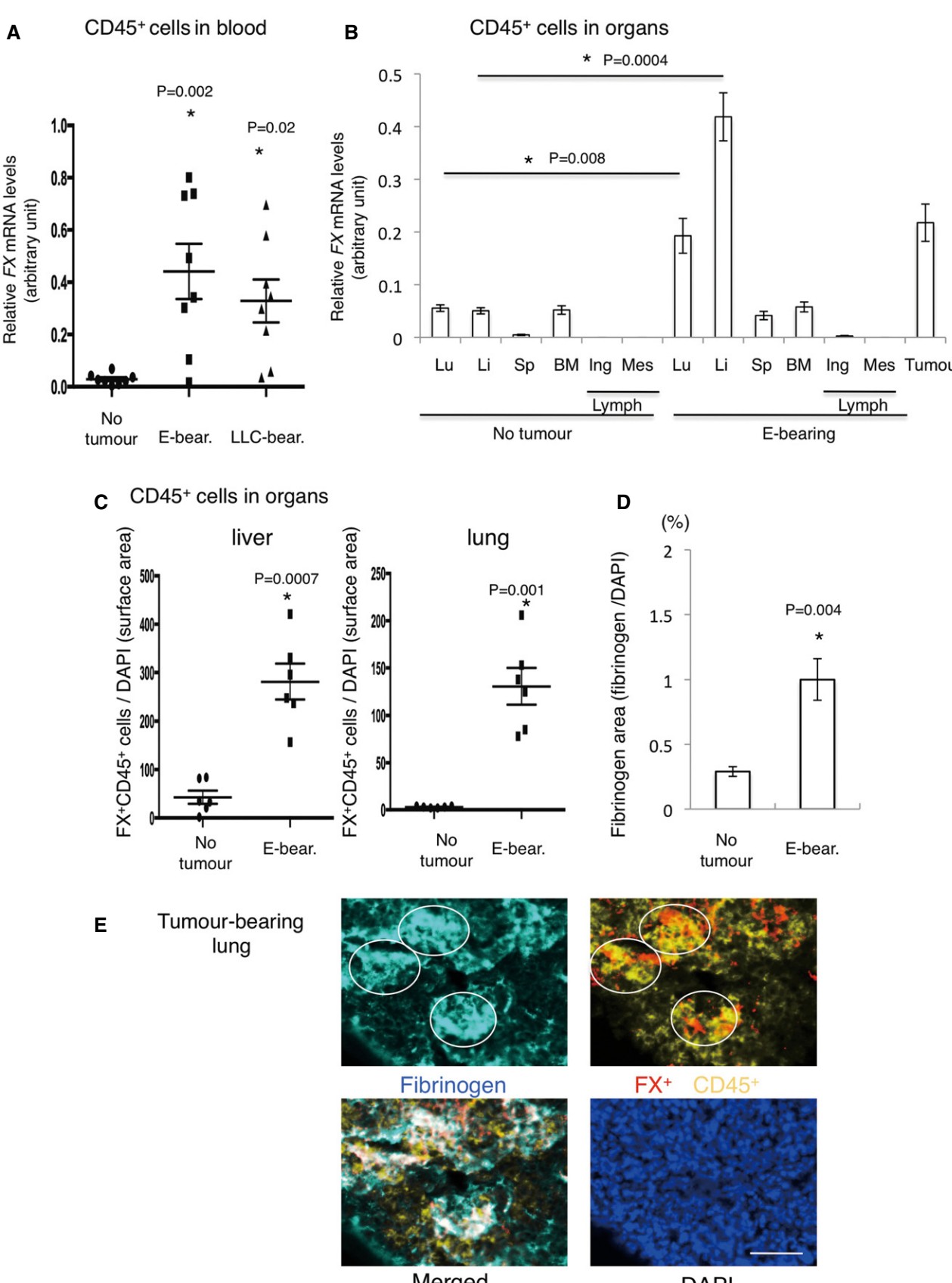

**Figure 1.**

Figure 2.

◀

**Figure 2.  *In vivo* tracking of HepELs in a primary tumour-stimulated mouse.**

A    An experimental tracing model of CD45$^+$ leucocytes in "KikGR" mice using a photoconversion system. Colour conversion from KikGR green-to-KikGR red occurred in liver cells upon violet light irradiation. In the tumour-bearing- or tumour-conditioned media (TCM)-stimulated KikGR mice, and the cells later moved into the lungs. The KikGR red cells, obtained from TCM-stimulated liver and lungs, were isolated by a cell sorter and CD45 microbeads, and these purified cells were used for microarray screening.

B    KikGR red cells were detected in the TCM-stimulated liver and lungs after liver exposure to violet light (arrow). Images taken from animals with no light exposure were also shown (scale bar, 100 μm).

C    Flow cytometric quantifications of photoconverted HepELs in TCM (three times)-stimulated KikGR mouse liver and lungs. Cells were isolated 72 h after photoconversion. Ratio was calculated as the number of photoconverted cells (KikGR red) observed in the region (gated in Fig EV1) in comparison with the number of liver or lung cells pre-sorted with CD45-beads. Shown are averages (*N* = 5) with SEM and Welch's *t*-test.

D    Surface marker analyses of photoconverted KikGR cells. Vertical axes represent ratio of each marker+ cell/photoconverted KikGR cell. Shown are averages (conCM, control: *N* = 4, TCM: *N* = 9) with SEM and Welch's *t*-test.

E    Representative immunostaining images of FX expression in NK1.1$^+$, CD11c$^+$, B220$^+$ and CD4$^+$ cells in tumour-bearing lungs (scale bar, 10 μm).

After 72 h, the photoconverted KikGR red cells continued to produce KikGR green resulting in shifting upper-left direction in the dot plot (see arrow in Fig EV1D). The ratio of photoconverted cells in the lungs of unstimulated mice was low; however, it became tangible in the case of TCM-stimulated KikGR mice (Appendix Fig S5A and B). We further analysed the ratios of photoconverted KikGR cells in the liver and lung by using TCM-stimulated KikGR mice. The ratios of photoconverted CD45$^+$ cells per total CD45$^+$ cells in liver were higher than those in lung (*N* = 5, liver, 0.52 ± 0.08%, lung, 0.12 ± 0.02% (mean ± SEM) in Fig 2C). Our results demonstrated that a part of liver CD45$^+$ cells that underwent photoconversion were diverted to the lung (Fig 2C and Appendix Fig S5B). To determine the cell surface markers expressed in lung HepELs, we conducted flow cytometric analyses using antibodies for CD45, NK1.1, CD4, CD8, CD11b, CD11c and B220 using spleen and lungs (Appendix Figs S3 and S4). First, it should be emphasized that very few photoconverted CD45-negative cells were observed in the lungs (Fig EV1C and E). Second, TCM stimulation increased the populations of CD4$^+$CD45$^+$, CD8$^+$CD45$^+$, NK1.1$^+$CD45$^+$, CD11c$^+$CD45$^+$ and B220$^+$CD45$^+$ compared with conCM (control culture medium without tumour cells) stimulation (Figs 2D and EV1C and E). Since we confirmed that the primary tumours induced the relocation of CD4$^+$CD45$^+$, CD8$^+$CD45$^+$, NK1.1$^+$CD45$^+$, CD11c$^+$CD45$^+$ and B220$^+$CD45$^+$HepELs in the lungs, we examined whether these cells expressed FX. The FX signal was immunohistochemically detected in CD4$^+$CD45$^+$, NK1.1$^+$CD45$^+$, CD11c$^+$CD45$^+$ and B220$^+$CD45$^+$ cells, indicating that FX$^+$ HepELs are a mixed population of mononuclear cells (MNCs) (Fig 2E). To clarify which type of cell relocated to the pre-metastatic lung, we further examined photoconverted HepELs using candidate cell markers such as CD4, NK1.1, CD11c and B220. We found that about 1% of B220$^+$CD11c$^+$NK1.1$^+$ leucocytes emerged in TCM-stimulated lungs (Fig 3A and B). On the other hand, control CM (conCM) did not induce B220$^+$CD11c$^+$NK1.1$^+$ cells in the lung (Fig 3A). As shown in Figs 3B and EV2, the relocated photoconverted HepELs in TCM-stimulated lungs were confirmed to be B220$^+$CD11c$^+$NK1.1$^+$ leucocytes. In examination of the relocated HepELs, TCRβ$^+$NK1.1$^-$ T cell mobilization was not observed in TCM-stimulated lungs (Appendix Fig S6). We could not clarify whether B220$^+$CD11c$^+$NK1.1$^+$ cells were also NK1.1$^+$TCRβdim NKT cells in this assay system (Appendix Fig S6). B220$^+$CD11c$^+$NK1.1$^+$ cells in various organs such as lung, liver, peripheral blood, bone marrow, lymph node and the primary tumour were investigated. We collected samples 2, 7 and 14 days after the tumour cell implantation; their approximate tumour sizes were 0 mm (2 days),

3 mm (7 days) and 10 mm (14 days) in diameter, respectively. Among them, the FX expression levels in B220$^+$CD11c$^+$NK1.1$^+$ cells isolated from the liver of 3 mm tumour-bearing mice were remarkably high (Appendix Fig S7, upper panel). We would like to note that the FX expressions in the cells derived from the lung and tumour tissues in 10 mm tumour-bearing mice were also observed (Appendix Fig S7, upper panel).

## B220$^+$CD11c$^+$NK1.1$^+$ HepELs attack tumour cells via IFNγ

It has been reported that B220$^+$CD11c$^+$NK1.1$^+$ cells were identified as a subset of natural killer (NK) cells in lymphoid organs (Blasius *et al*, 2007). NK cells are prototypic innate lymphoid cells endowed with potent cytolytic function, which provides a host defence against tumours and microbial infections (Robinette *et al*, 2015; Morvan & Lanier, 2016). In addition, NK cell-type cytotoxic capacities of CD3$^-$NK1.1$^+$ cells were reduced by hypoxic primary tumour-derived factors in the pre-metastatic niche (Sceneay *et al*, 2012). To investigate the nature of B220$^+$CD11c$^+$NK1.1$^+$ cells against metastatic tumour cells, we examined the IFNγ secretion capacity and NK cell cytotoxic activity. For IFNγ secretion, we compared organ-specific microenvironments such as liver and lungs. The IFNγ in liver B220$^+$CD11c$^+$NK1.1$^+$ cells was kept at a moderate level, but was markedly induced by conditioned media (CM) cultured with LLC-bearing lungs (Fig 4A and B left). In contrast, IFNγ induction in spleen with B220$^+$ CD11c$^+$ NK1.1$^+$ cells was prominent by both liver-CM and lung-CM (Fig 4B, right). To determine whether NK cell-mediating cytotoxic activity is dependent on tumour-stimulating organ education, we co-cultured LLC cells with B220$^+$CD11c$^+$NK1.1$^+$ cells derived from liver or lung in LLC-bearing mice. In accordance with IFNγ induction, lung-CM-primed B220$^+$CD11c$^+$NK1.1$^+$ cells clearly attacked co-existing tumour cells (Fig 4C–E).

## Differential expression patterns of FX$^+$CD45$^+$ cells between liver and lungs of tumour-bearing mice

To better understand the molecular mechanisms mediating HepEL relocalization from the liver to the lungs in response to tumour progression, we isolated photoconverted HepELs from the liver and lungs of TCM-treated mice using a cell sorter (see Materials and Methods, "Isolation of CD45$^+$ cells and B220$^+$CD11c$^+$NK1.1$^+$ cells"). Gene expression between the two cell populations was assessed by microarray analysis (Appendix Table S2). Interestingly, HepELs remaining in the liver displayed increased Vtn (Seiffert *et al*, 1994) expression compared with those that migrated to the lungs

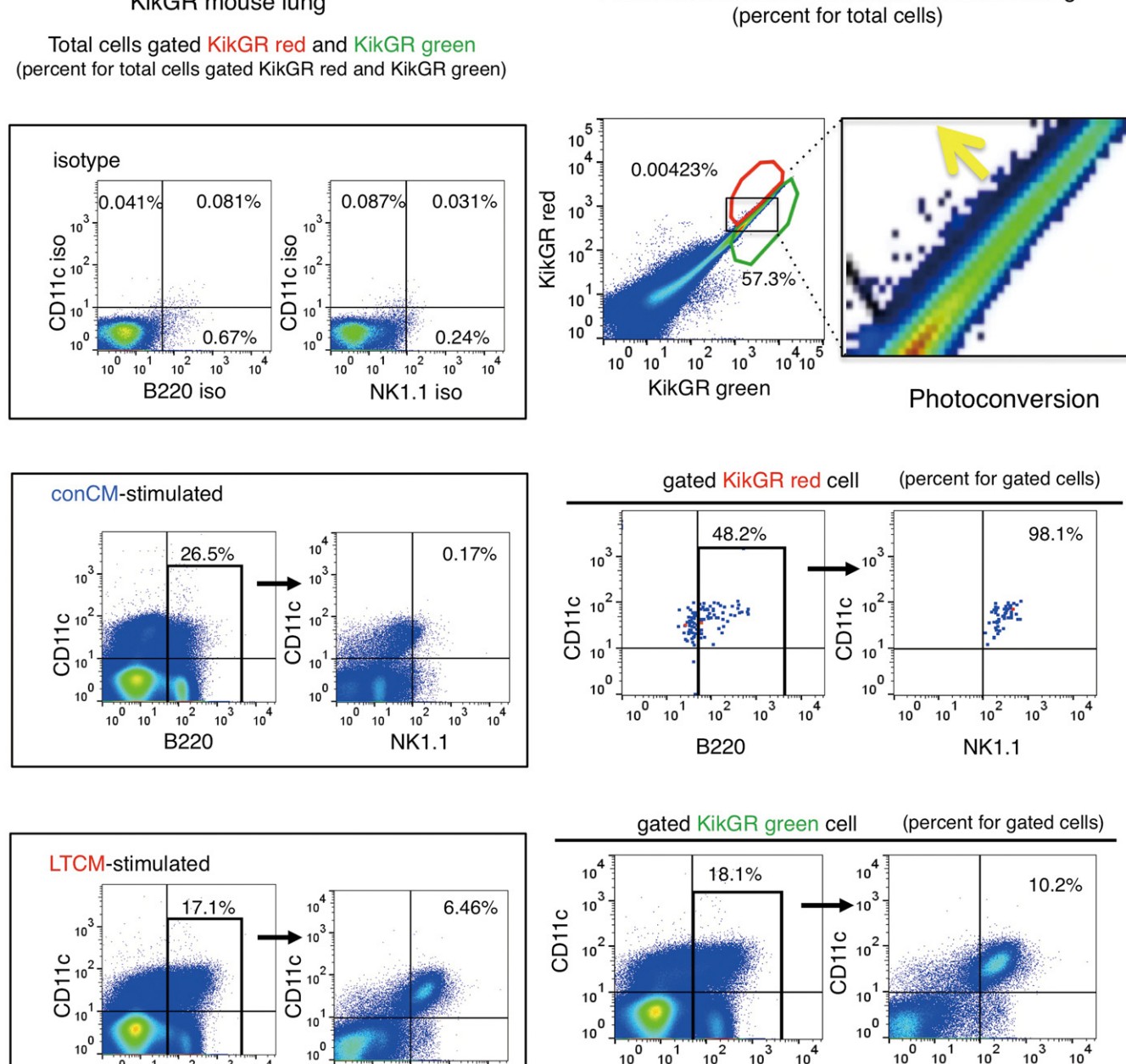

**Figure 3.  Relocation of B220⁺CD11c⁺NK1.1⁺HepELs in pre-metastatic lungs.**

A   Flow cytometric analysis of B220⁺CD11c⁺NK1.1⁺ cells in the lung of TCM-stimulated mice. Values in the dot plots present ratios for total cells gated KikGR red and KikGR green cell.

B   Representative flow cytometric analyses of the relocation of B220⁺CD11c⁺NK1.1⁺ cells in lungs that were photoconverted in TCM-stimulated KikGR mouse liver (gated in red polygonal region). Percent is shown for total cells (upper panel) and for gated KikGR red or KikGR green cells (middle and lower panels). Region where the photoconverted cells locate is magnified in the right. Yellow arrow shows the photoconversion direction. Three independent experiments. LTCM stands for TCM derived from LLC.

(Appendix Table S2, right). In contrast, lung HepELs showed a higher expression of thrombospondin (TSP) (*THBS1*) (Good *et al*, 1990; Watnick *et al*, 2014) (Appendix Table S2, left). Notably, both Vtn and TSP have been reported to bind fibrinogen (Panetti *et al*,

1999; Podor *et al*, 2002), which is prevalent in the focal areas of pre-metastatic lungs. We attempted to visualize Vtn/TSP expression pattern in autopsy samples from cancer patients. First, we pathologically examined the liver and lung tissue to identify regions devoid

**A**

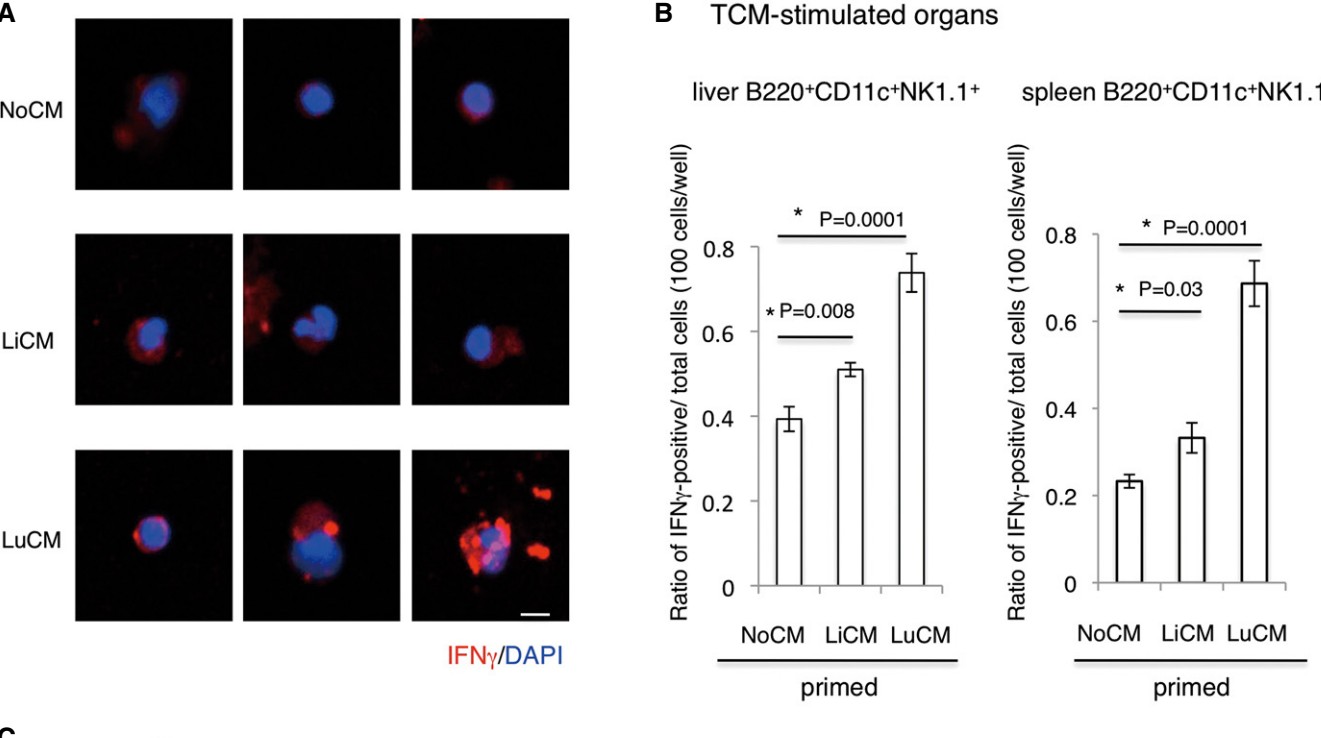

NoCM

LiCM

LuCM

IFNγ/DAPI

**B**    TCM-stimulated organs

liver B220⁺CD11c⁺NK1.1⁺          spleen B220⁺CD11c⁺NK1.1⁺

**C**

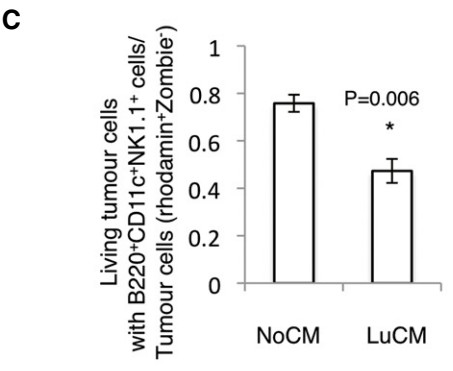

**D**

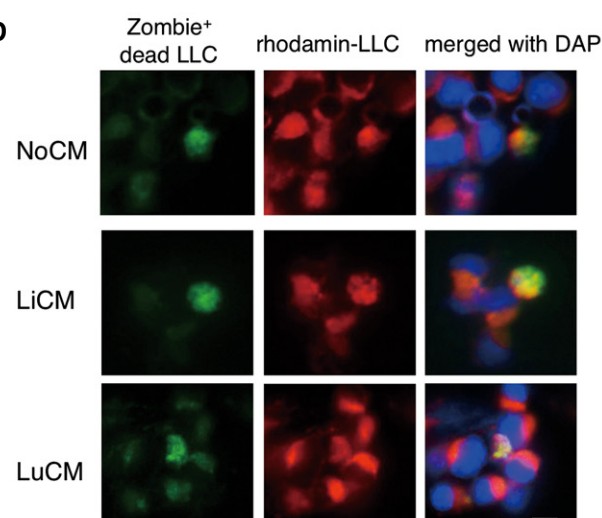

**E**    TCM-stimulated organs

liver B220⁺CD11c⁺NK1.1⁺     spleen B220⁺CD11c⁺NK1.1⁺

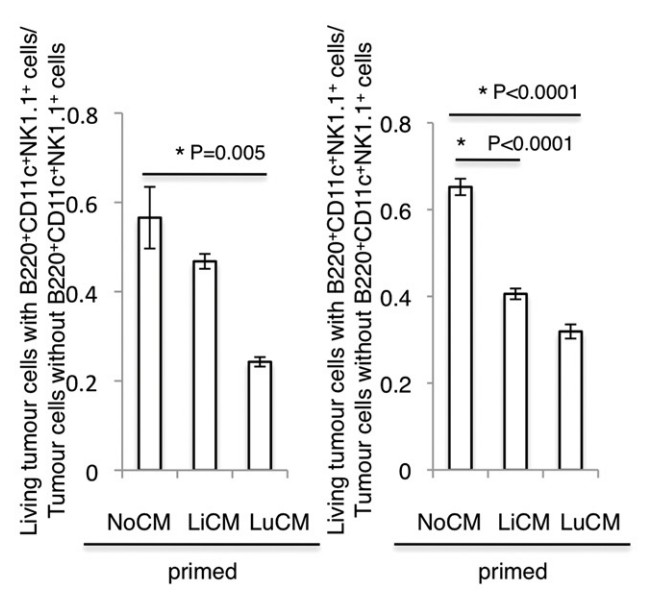

**Figure 4.**

**Figure 4. IFNγ induction and anti-tumour activity of B220$^+$CD11c$^+$NK1.1$^+$HepELs.**

A Representative photographs of IFNγ expression in B220$^+$CD11c$^+$NK1.1$^+$ cells derived from TCM-stimulating mouse liver. These cells were cultured with LiCM or LuCM that had been prepared by culture with tumour-bearing- liver or lungs, respectively. NoCM stands for CM without tissues, and it was used as control (scale bar, 10 μm).

B Immunohistochemical quantifications of IFNγ in B220$^+$CD11c$^+$NK1.1$^+$ cells. Cells derived from liver and spleen were primed with LiCM or LuCM. Shown are averages ($N = 6$/group) with SEM and one-way ANOVA.

C Flow cytometric analysis of living Zombie$^-$ tumour cells with B220$^+$CD11c$^+$NK1.1$^+$ cells that were primed by NoCM or LuCM. NoCM: $N = 3$, LuCM: $N = 4$. Shown are averages with SEM. Welch's $t$-test. Rhodamine$^+$ tumour cells were counted as viable cells.

D Representative immunohistochemical stainings of rhodamine$^+$ (red) Zombie$^+$ (green) tumour cells (scale bar, 10 μm).

E Ratios of living tumour cells after co-culture with B220$^+$CD11c$^+$NK1.1$^+$ cells that had been primed with lung- or liver-CM. Shown are averages ($N = 6$/group) with SEM and one-way ANOVA.

of tumour metastasis, atelectasis or inflammation and then performed immunohistochemistry to quantify the presence of Vtn$^+$FX$^+$CD45$^+$ and TSP$^+$FX$^+$CD45$^+$ cells in the liver and lungs of non-cancer and cancer patients. We found that Vtn expression in CD45$^+$ cells was higher in the liver than in the lung tissue from the same patient (Fig EV3A). In contrast, TSP expression was upregulated in CD45$^+$ cells in the lung, but not in the liver (Fig EV3A). A unique pattern of Vtn-TSP signal intensities in FX$^+$CD45$^+$ cells was seen in the liver and lungs of non-cancer and cancer patients (Fig EV3B). Furthermore, the increased number of FX$^+$CD45$^+$ cells was observed in cancer patients (Fig EV3B).

## B220$^+$CD11c$^+$NK1.1$^+$ HepELs eliminate fibrinogen deposition via FX

Our microarray data indicate that coagulation factor 5 and factor 13 as well as factor 10 (FX) were packed in peripheral blood leucocytes in tumour-bearing mouse (Appendix Table S1: GSE76506). Among them, FX was upregulated in tumour-bearing mice. We set up a coagulation assay system to measure prothrombin time (PT) of B220$^+$CD11c$^+$NK1.1$^+$ cells derived from tumour-bearing mice. To determine PT, we recorded absorbance at 671 nm after mixing HemosIL RecombiPlasTin with samples. In our assay, 50 mg/dl of purified fibrinogen showed 10 s of PT. Then, we examined the effect of B220$^+$CD11c$^+$NK1.1$^+$ cells. The B220$^+$CD11c$^+$NK1.1$^+$ cells ($5 \times 10^3$ cells) derived from the lung or liver in tumour-bearing mouse showed PTs of 176 s (lung, $n = 3$) and 190 s (liver, $n = 6$), respectively. These data imply that the B220$^+$CD11c$^+$NK1.1$^+$ cells play a role in coagulation cascade.

Next, we investigated whether the relocated B220$^+$CD11c$^+$ NK1.1$^+$ HepELs have FX-dependent function in pre-metastatic niches. We have shown that focal hyperpermeable regions in pre-metastatic lungs serve as a preferable soil for circulating metastatic

tumour cells (Hiratsuka et al, 2011, 2013). Therefore, regional fibrinogen deposition may be a consequence of hyperpermeability, which is a metastatic scaffold for BMDC mobilization and activation of adhesion molecules in lung endothelial cells. Fibrinogen is principally synthesized in the liver as a plasma glycoprotein composed of two sets of three different polypeptide chains (Aα, Bβ and γ) (Takezawa et al, 2013). We detected that primary tumours stimulated fibrinogen production, particularly the γ chain, in tumour-bearing livers (Fig 5A), as well as increases of fibrinogen deposition in tumour-bearing lungs (Fig 1D). Given that fibrinogen is a potential binding partner for Vtn and TSP (Panetti et al, 1999; Podor et al, 2002), it would be worth investigating the interactions between fibrinogen and Vtn or TSP in tumour-bearing organs. In addition, we hypothesized that mobilizing B220$^+$CD11c$^+$NK1.1$^+$ HepELs eliminate lung fibrinogen deposition via FX (Fig 5B).

To decipher the role of FX in HepELs, we used viable FX-deficient mice, with only 5.5% of the FX activity observed in wild-type mice (Tai et al, 2008). We first tested the interaction between fibrinogen and Vtn or TSP in FX$^{+/-}$ and FX$^{-/-}$ mice. Co-immunoprecipitation studies using an anti-fibrinogen antibody showed that less Vtn and TSP were bound to fibrinogen in tumour-bearing-FX$^{-/-}$ mice compared with tumour-bearing- FX$^{+/+}$ or FX$^{+/-}$ controls (Fig 5C). We detected no difference of binding between fibrinogen and Vtn or TSP among those genotypes (see legend in Fig 5C). Next, we investigated whether FX-mediated coagulation affected the degree to which HepELs accumulated in the focal areas with fibrinogen in tumour-bearing lungs. To this end, the removal of fibrinogen was evaluated after an injection of HepELs from tumour-bearing wild-type mice. To determine the appropriate number of HepELs, we injected HepELs in a dilution series (Fig 5D). We found that $1 \times 10^6$–$5 \times 10^5$ HepELs derived from tumour-bearing mice were able to eliminate the existing fibrinogen deposition. We then examined FX-dependent fibrinogen elimination with the application of

**Figure 5. Fibrinogen elimination ability of Vtn$^+$HepELs via FX.**

A Western blotting of fibrinogen in tumour-bearing and no tumour-bearing livers.

B Scheme of the coagulation cascade related to FX and fibrinogen. Elimination of lung fibrinogen by activated FX (FXa) in B220$^+$CD11c$^+$NK1.1$^+$HepELs.

C IP—Western blot analysis shows protein–protein interactions between fibrinogen and Vtn/TSP.

D Quantification of fibrinogen deposition areas after injection of CD45$^+$ cells derived from the liver in tumour-bearing mice. Shown are averages ($N = 7$/group) with SEM. One-way ANOVA.

E Quantifications of fibrinogen deposition areas in TCM-stimulating wild-type mouse lungs after injection of Vtn$^+$ HepELs. Vtn$^+$ cells were isolated from E0771-bearing wild-type or FX$^{-/-}$ mouse liver by using anti-Vtn beads ($1 \times 10^6$ cells/mouse). Shown are averages ($N = 5$/group, 3 fields/sample) with SEM. Welch's $t$-test.

F Separation of Vtn$^+$CD45$^+$ HepELs from tumour-bearing FX$^{+/-}$ and FX$^{-/-}$ mouse livers by a cell sorter (Moflo Astrios). Sorted Vtn$^+$CD45$^+$HepELs cells obtained from a littermate (3 animals each) were pooled, and $8 \times 10^3$ Vtn$^+$CD45$^+$HepELs cells were i.v. injected into a tumour-bearing mouse. Right graph shows fibrinogen depositions in the lung. Shown are averages ($N = 3$/group, 9 fields/sample) with SEM. 1-way ANOVA.

Source data are available online for this figure.

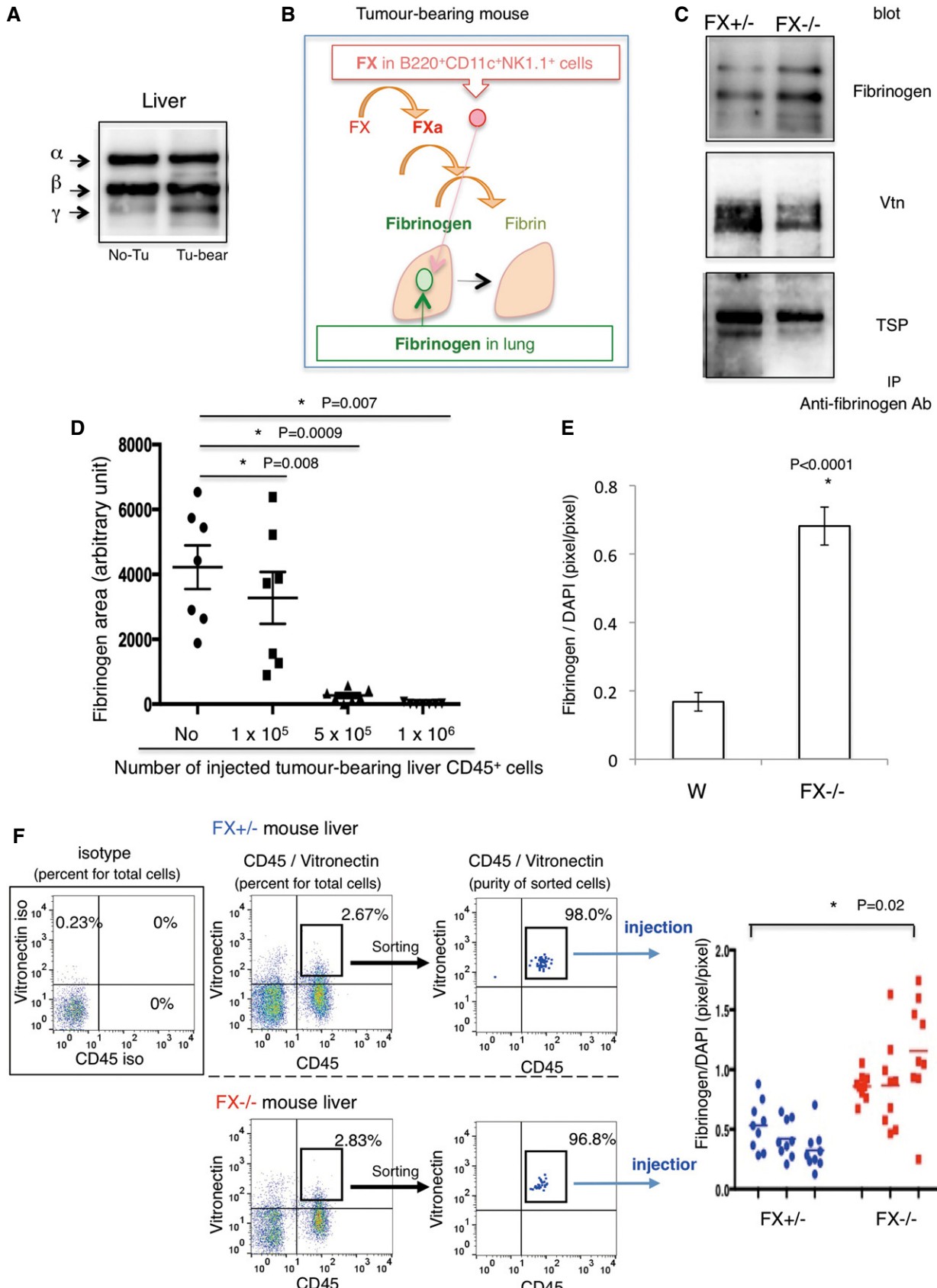

**Figure 5.**

tumour-bearing Vtn$^+$HepELs. To address the fibrinogen clearance ability of Vtn$^+$HepELs, we injected Vtn$^+$HepELs ($1 \times 10^6$ cells) isolated from tumour-bearing wild-type (FX$^{+/+}$) or FX$^{-/-}$ mice. Application of FX$^{+/+}$Vtn$^+$HepELs decreased more fibrinogen in tumour-bearing mouse lungs than did application of FX$^{-/-}$ Vtn$^+$HepELs (Fig 5E). Although the cell number used for the injection was reduced to $8 \times 10^3$ cells/mouse, fibrinogen elimination by FX-dependent HepELs was observed (Fig 5F). To search the relocated HepELs in the FX$^{-/-}$ mouse background, we examined whether CD45$^+$HepELs derived from FX$^{+/+}$ and FX$^{-/-}$mouse eliminated fibrinogen deposition in lungs in tumour-bearing FX$^{-/-}$ mice. We observed twofold to fourfold higher deposition of fibrinogen in tumour-bearing FX$^{-/-}$ mouse lungs than that of tumour-bearing wild-type mouse because of abrogation of effective fibrinogen clearance (Appendix Fig S8, left). In spite of abundant fibrinogen deposition, FX$^{+/+}$CD45$^+$HepELs more significantly eliminated fibrinogen deposition than FX$^{-/-}$CD45$^+$HepELs (Appendix Fig S8, right). Metastatic tumour cell homing was reduced by application of FX$^{+/+}$ CD45$^+$HepELs or FX$^{+/-}$NK1.1$^+$HepELs compared with the corresponding HepELs from FX$^{-/-}$ livers in tumour-bearing FX$^{-/-}$ mouse lungs (Appendix Fig S9). These data let us examine the fibrinogen elimination by B220$^+$CD11c$^+$NK1.1$^+$HepELs in the tumour-bearing wild-type mouse background, since fibrinogen deposition in tumour-bearing FX$^{-/-}$ mice was more prominent than that in tumour-bearing wild-type mice.

Finally, we carried out fibrinogen elimination assays using B220$^+$CD11c$^+$NK1.1$^+$HepELs. Approximately 1% of B220$^+$CD11c$^+$ NK1.1$^+$ HepELs were found among total tumour-bearing lung CD45$^+$ leucocytes (Fig 3A and B), and $1 \times 10^6$ of liver CD45$^+$ leucocytes eliminated lung fibrinogen (Fig 5D). Thus, we selected an injection cell number of $3 \times 10^4$ of B220$^+$CD11c$^+$NK1.1$^+$HepELs expecting that this may erase fibrinogen in tumour-bearing mice. The B220$^+$CD11c$^+$NK1.1$^+$HepELs derived from tumour-bearing FX$^{+/-}$ mice significantly reduced fibrinogen deposition (Fig 6A and B). In contrast, those from tumour-bearing FX$^{-/-}$ mice lacked this capacity (Fig 6A and B).

### Liver-primed B220$^+$CD11c$^+$NK1.1$^+$HepELs protect against metastasis

To generate B220$^+$CD11c$^+$NK1.1$^+$ HepELs, which were primed by the tumour-bearing liver microenvironment *in vitro*, we set up an assay system illustrated in Fig 7A. In the system, B220$^+$CD11c$^+$ NK1.1$^+$ HepELs from tumour-bearing mice were seeded in the lower well, and various diced tissues were placed in the upper culture insert. By using this system, we investigated whether HepELs are affected by humoral factors derived from the various tissues. Vtn and FX expressions in B220$^+$CD11c$^+$NK1.1$^+$HepELs were reduced in the control condition (Fig 7B, NoCM), whereas tumour-bearing liver tissues upregulated expression (Fig 7B). However, when these cells were stimulated by lung-conditioned media (CM) from tumour-bearing mice, TSP expression was upregulated compared with counterparts co-cultured with liver-CM (Fig 7B). Next, we examined whether any molecule derived from primary tumours induces FX expression in HepELs in the pre-metastatic phase. Because it has been reported that the tumour-derived factors such as CCL2, SDF1, IL6, TNFα, VEGF, G-CSF, TGFβ, and CXCL1 function in the pre-metastatic phase (McAllister & Weinberg,

2014; Wang *et al*, 2017), we applied those factors to HepELs in the tumour-bearing mouse liver *in vitro* (Fig 7C). We found out that CCL2 and CXCL1 strongly induced FX in HepELs. These data indicate involvement of tumour-derived factors in the regulation of HepELs (Fig 7C).

We examined the contribution of Vtn in B220$^+$CD11c$^+$ NK1.1$^+$HepELs to cell adhesion through fibrinogen by using the adhesion assay (Fig 7A). Rhodamine-labelled B220$^+$CD11c$^+$ NK1.1$^+$HepELs, educated by the tumour-bearing liver, promoted the attachment of these cells to fibrinogen-coated plates (Fig 7D). The addition of anti-Vtn antibody reduced the number of attached cells (Fig 7D). We focused on TSP as a ligand molecule of fibrinogen because TSP was upregulated in lung HepELs but not in liver HepELs (Appendix Table S2). Our results exhibit that anti-Vtn Ab inhibited binding of lung HepELs to a fibrinogen-coated plate. Similarly, neutralizing anti-TSP Ab blocked the binding of lung HepELs (Fig 7E). Then, we examined binding abilities of HepELs to other ECM such as collagen I and fibronectin (FN). Our data show that HepELs were able to attach to collagen I or FN although their affinities were not as high as that to fibrinogen (Fig 7E). Moreover, the HepEL-FN/collagen I interactions were Vtn/TSP independent.

Finally, we examined the effect of B220$^+$CD11c$^+$NK1.1$^+$HepEL-mediated fibrinogen removal with and without FX (method in Fig 8A). Notably, the homing of metastatic tumour cells was reduced upon the elimination of fibrinogen deposits in FX$^{+/+}$ B220$^+$CD11c$^+$NK1.1$^+$HepEL-treated tumour-bearing mice (Fig 8B). In contrast, the administration of FX$^{-/-}$ B220$^+$CD11c$^+$NK1.1$^+$HepELs failed to suppress subsequent tumour cell homing (Fig 8B).

We tried to establish an activated factor X-overexpression system (FX-OE) in B220$^+$CD11c$^+$NK1.1$^+$ cells. Briefly, biotinylated polyethylene glycol (PEG)-lipid and biotinylated recombinant FX were combined with neutralized avidin to tether FX to B220$^+$CD11c$^+$ NK1.1$^+$ cells. Then, we characterized the FX-tethered B220$^+$ CD11c$^+$NK1.1$^+$ cells in a clotting assay to measure their prothrombin time (PT). The $2.5 \times 10^3$ of FX-OE-B220$^+$CD11c$^+$NK1.1$^+$ cells (FX-OE HepELs) derived from tumour-bearing mouse lungs showed shorter PT than the HepELs without recombinant FX (56 s ($n = 3$) and 193 s ($n = 3$), respectively). Next, we compared these two cells in the metastasis assay. The number of metastatic rhodamine-labelled tumour cells was decreased in tumour-bearing lungs after injection of lung HepELs, and FX-OE HepELs enhanced the inhibitory activity of HepELs (Fig 8C).

In summary, B220$^+$CD11c$^+$NK1.1$^+$ cells primed by the tumour-stimulated liver have an anti-metastatic function during primary tumour progression. In the pre-metastatic phase, these cells relocate from liver to lung and attach to fibrinogen-rich areas via Vtn. Simultaneously, these cells eliminate fibrinogen deposition resulting in the suppression of pre-metastatic soil. In the metastatic phase, B220$^+$CD11c$^+$NK1.1$^+$HepELs with high IFNγ activity attack migrating metastatic tumour cells in the lung (Fig 8D).

## Discussion

In some organs, the local microenvironment can be stimulated by a distant primary tumour prior to metastasis, producing conditions referred to as pre-metastatic soil (=niche) (Hiratsuka *et al*, 2002; Kaplan *et al*, 2005; Sceneay *et al*, 2013; McAllister & Weinberg,

**A** FX-/- B220⁺CD11c⁺NK1.1⁺ cells

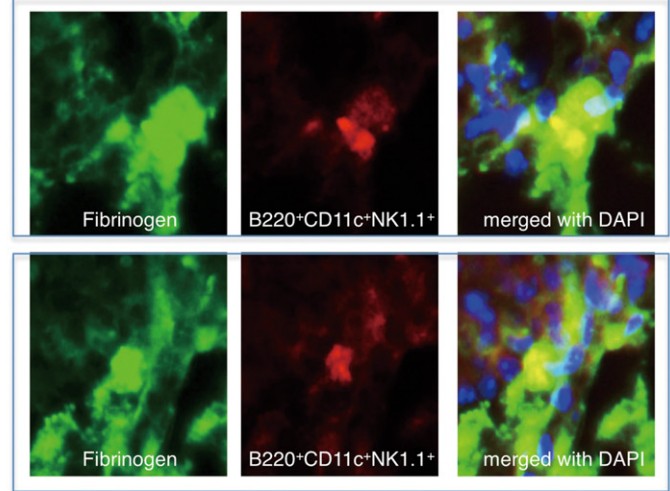

FX+/- B220⁺CD11c⁺NK1.1⁺ cells

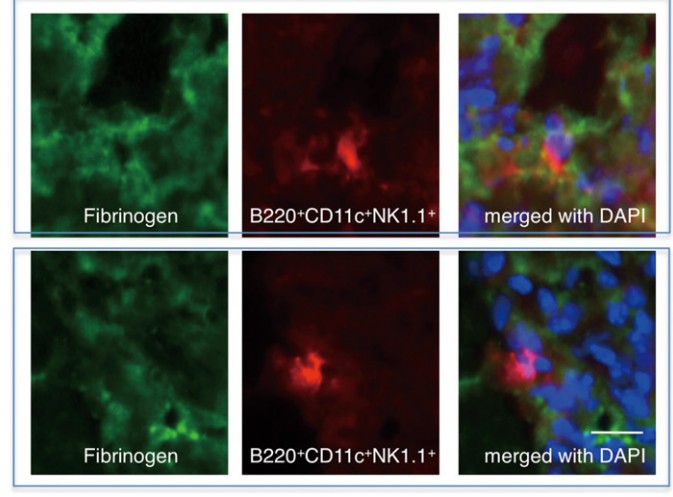

**B**

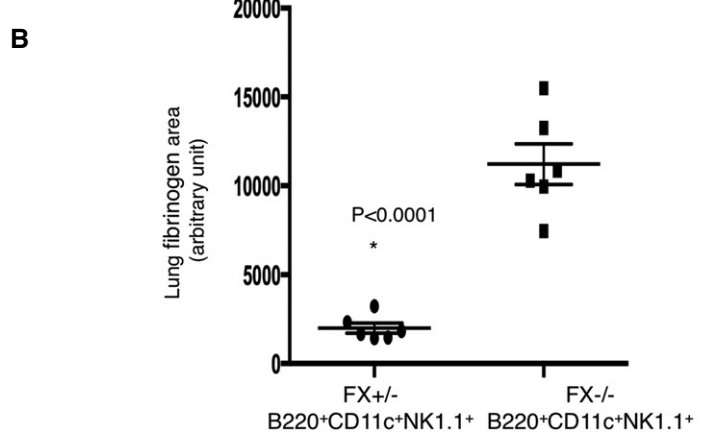

**Figure 6. Fibrinogen elimination of B220⁺CD11c⁺NK1.1⁺HepELs via FX in pre-metastatic lungs.**

A  Representative immunostaining of fibrinogen depositions in tumour-bearing mouse lungs received an injection of rhodamine-labelled B220⁺CD11c⁺NK1.1⁺HepELs derived from tumour-bearing- FX$^{+/-}$ or FX$^{-/-}$ mice (scale bar, 20 μm).

B  Immunohistochemical quantifications of fibrinogen depositions in tumour-bearing mouse lung after an injection of B220⁺CD11c⁺NK1.1⁺ cells derived from tumour-bearing- FX$^{+/-}$ or FX$^{-/-}$ mouse liver. Sorted B220⁺CD11c⁺NK1.1⁺ cells were prepared from three mice per each genotype and were i.v. injected. Shown are averages (N = 6/group, 4 sections/mouse) with SEM and Welch's *t*-test.

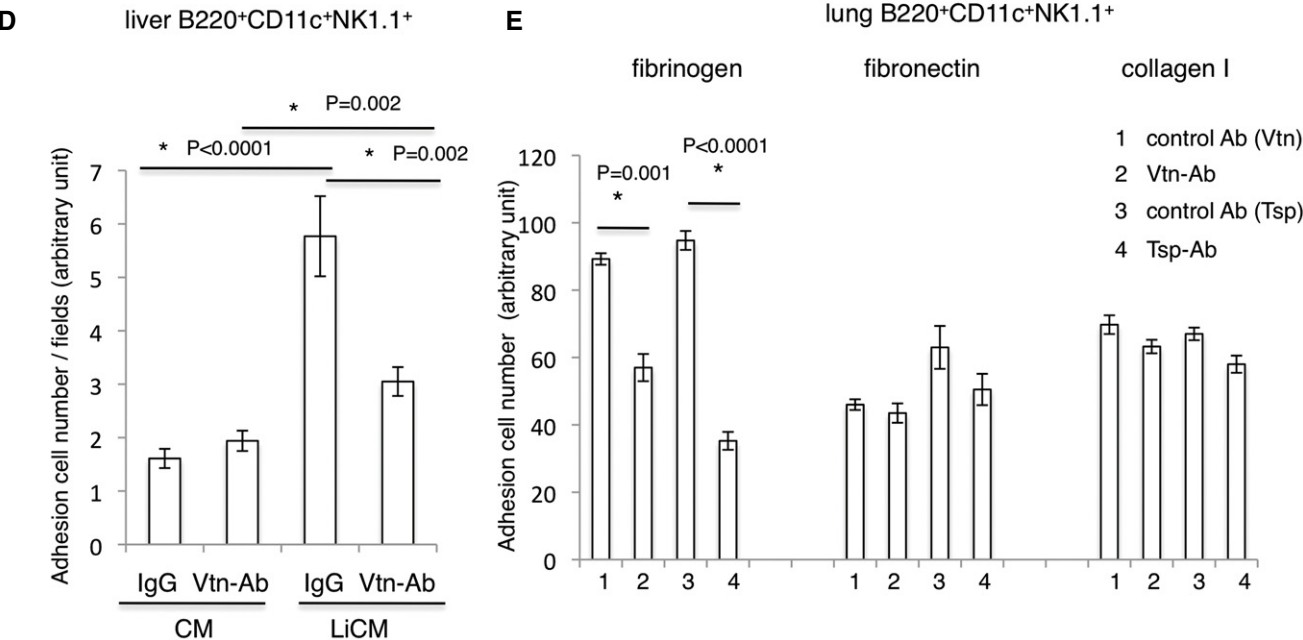

Figure 7.

**Figure 7.  Vtn-dependent B220⁺CD11c⁺NK1.1⁺HepEL attachment to fibrinogen.**

A   Adhesion assay of B220$^+$CD11c$^+$NK1.1$^+$ cells to fibrinogen-coated plates. The rhodamine-labelled liver B220$^+$CD11c$^+$NK1.1$^+$ cells were separately cultured with TCM-stimulating liver tissues and seeded on a fibrinogen-coated plate.

B   Relative mRNA levels of *FX*, *Vtn* and *TSP1* in liver B220$^+$CD11c$^+$NK1.1$^+$ cells stimulated with LiCM or LuCM. Shown are averages (*N* = 6/group) with SEM and one-way ANOVA.

C   Relative mRNA levels of *FX* in liver B220$^+$CD11c$^+$NK1.1$^+$ cells stimulated with various factors. Shown are averages (*N* = 4) with SEM and one-way ANOVA.

D   *In vitro* adhesion assay. Liver B220$^+$CD11c$^+$NK1.1$^+$ cells were seeded on a fibrinogen plate. Cells attached to the plate were counted. Twenty μg/ml of anti-Vtn Ab or isotype control IgG was used to block the adhesion. Shown are averages (*N* = 6/group) with SEM and one-way ANOVA.

E   Lung B220$^+$CD11c$^+$NK1.1$^+$ cells were seeded on ECM-coated (fibrinogen, fibronectin or collagen I) plates. Shown are averages (*N* = 5/group) with SEM and one-way ANOVA.

2014). Metastatic mobilization is considered to be generated through a triangular interplay among primary tumours, the pre-metastatic tissue and the BM (Wels *et al*, 2008). Although the tumour-induced systemic environment likely affects metastases, the precise role of metastatic and non-metastatic organ interplay during the pre-metastatic phase has yet to be fully elucidated. Studies of the pre-metastatic phase have primarily focused on the lungs. Pre-metastatic lungs contain a variety of tissue resident cells, such as endothelial cells, resident macrophages and club cells, as well as infiltrating cells such as peripheral blood cells and BMDCs. Thus, the pre-metastatic soil may contain a complex mix of factors either inhibit or promote metastasis. Various types of immune cells have been identified in the pre-metastatic soil. For example, tumour-entrained neutrophils (TENs) with CD11b$^+$ (Ly-6G)$^+$ inhibited metastatic seeding (Granot *et al*, 2011), while tumour-associated macrophages (TAMs), including metastasis-associated macrophages (MAMs) with CD11b$^+$, F4/80$^+$ and CSF-1R$^+$, enhanced tumour progression and metastasis (Qian & Pollard, 2010; Qian *et al*, 2015). CCL2 has been shown to be a critical mediator of the anti-metastatic entrainment of G-CSF-stimulated TENs (Granot *et al*, 2011). Nevertheless, CCL2-activated TAMs are associated with a poor prognosis in cancer (Qian & Pollard, 2010).

In this study, we showed that hepato-entrained B220$^+$CD11c$^+$NK1.1$^+$ cells expressing FX and Vtn migrated from the liver to the lungs during the pre-metastatic phase. This movement was accompanied by switching of the expression balance between Vtn and TSP, which was regulated by a distant primary tumour and associated with anti-metastasis via FX-mediated fibrinogen elimination. In addition, these cells had NK-like activity with IFNγ secretion in the case of metastatic niche.

The B220$^+$CD11c$^+$NK1.1$^+$HepELs derived from TCM-primed mouse attacked tumour cells *in vitro* (Fig 4C–E). Moreover, they were found in the primary tumour, and they became FX$^+$ in the later stage of tumour (Appendix Fig S7). These data suggest that HepELs potentially have anti-metastatic activity. However, given the fact that the HepELs population in tumour tissue decreased slightly in the primary tumour growth (Appendix Fig S7), the effect of HepELs against the primary tumour is limited. HepEL movement also declined during primary tumour progression. To obtain efficient anti-tumour activity, the relocation should be promoted and liver character, such as FX and Vtn expressions, should be boosted. We showed anti-metastatic results after the application of tumour-bearing liver-primed B220$^+$CD11c$^+$NK1.1$^+$ cells. B220$^+$CD11c$^+$NK1.1$^+$ cells were originally reported as NK cells with high expression of IFNγ in lymphoid organs (Blasius *et al*, 2007), and they may circulate in several organs during tumour progression. Regarding FX expression related to fibrinogen clearance in these cells, the tumour-stimulated liver appears to dominantly regulate it. Deciphering molecular mechanisms responsible for the FX induction in HepELs in a liver environment is ongoing. Based on our data, we speculate that FX expression in HepELs is regulated by multiple transcription factors such as Foxa1, Cebp-α and Rela (unpublished data). Regarding the role of HepELs, we showed that relocated B220$^+$CD11c$^+$NK1.1$^+$ cells eliminated fibrinogen deposition in pre-metastatic phase and killed metastatic tumour cells in post-metastatic phase. It has been reported that IFN-γ has dual opposite roles as anti-metastatic immune response and promotion of metastatic ability of tumour cells via activated nuclear factor κB (NF-κB) signalling pathway (Zhang *et al*, 2011; Xu *et al*, 2018). Among immune cells, it has been reported that IFN-γ is produced in CD3$^-$NK1.1$^+$, CD4$^+$ and CD8$^+$ T cells and CD3$^+$NK1.1$^+$ (NKT) cells. In this study, B220$^+$CD11c$^+$NK1.1$^+$ cells showed an anti-metastatic activity by eliminating fibrinogen in a FX-dependent manner in the pre-metastatic phase and by killing tumour cells in the post-metastatic phase. In addition, we observed that CD4$^+$HepELs, expressing FX and probably producing IFN-γ, promoted lung metastasis (unpublished data). Thus, some activated immune cells might support metastasis. Recently, organ-specific immune cells such as ILC (innate lymphoid

**Figure 8.  B220⁺CD11c⁺NK1.1⁺ cells primed with tumour-stimulating liver suppress metastasis via FX.**

A   The experimental scheme. B220$^+$CD11c$^+$NK1.1$^+$ cells derived from tumour-bearing mouse liver or lungs were primed with LiCM or tethered with recombinant FX and then injected into tumour-bearing or TCM-stimulated mice; 24 h after the injection, tumour cells were injected. Tissues were analysed 48 h after the tumour cell injection.

B   Tumour cell homing in the lungs pre-treated with LiCM-primed B220$^+$CD11c$^+$NK1.1$^+$ cells from tumour-bearing WT or FX$^{-/-}$ mouse liver. Representative photographs of homing of rhodamine-labelled tumour cells (upper, scale bar, 20 μm). Number of homing tumour cells are shown (lower). Shown are averages (*N* = 60 sections, 5/group, all field count/section, 12 sections/sample) with SEM and Welch's *t*-test.

C   Tumour cell homing in the lungs pre-treated with activated FX-overexpressed B220$^+$CD11c$^+$NK1.1$^+$ cells (FX-OE-HepELs) from tumour-bearing lungs. Number of homing tumour cells after an injection of lung HepELs or FX-OE-HepELs is shown. Shown here are averages (*N* = 24 sections, 4/group, all field count/section, 6 sections/sample) SEM and one-way ANOVA.

D   Model of B220$^+$CD11c$^+$NK1.1$^+$HepEL functions in the pre- and post- metastatic phases. In the pre-metastatic phase, tumour-stimulating liver induces FX expression in B220$^+$CD11c$^+$NK1.1$^+$ cells; then, the cells move into lungs to eliminate focal fibrinogens (left panel). In the post-metastatic phase, if circulating tumour cells reach the fibrinogen area, B220$^+$CD11c$^+$NK1.1$^+$HepELs attack tumour cells with IFNγ (right).

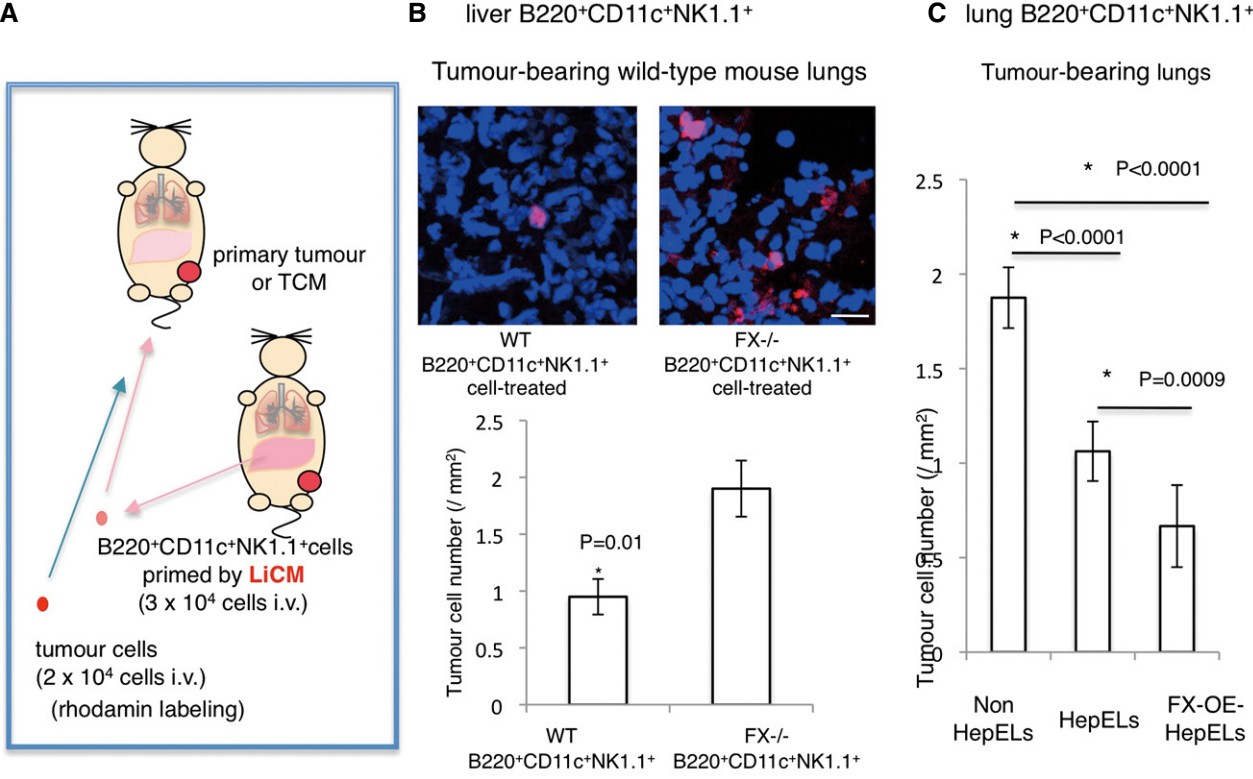

**A**

**B**   liver B220⁺CD11c⁺NK1.1⁺

**C**   lung B220⁺CD11c⁺NK1.1⁺

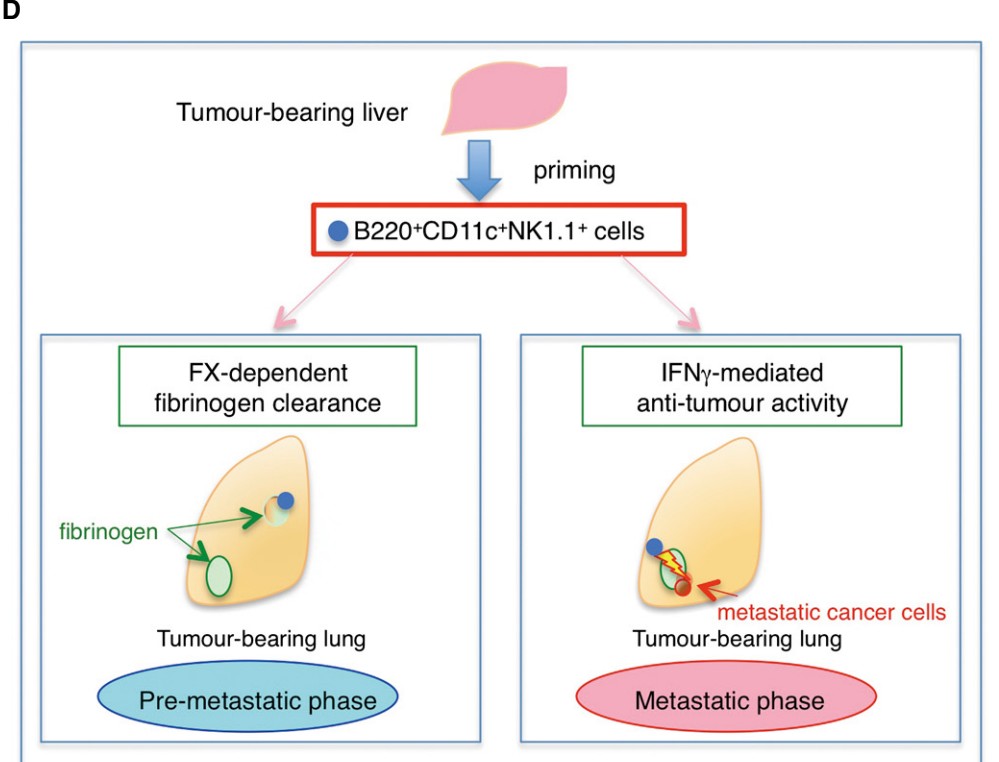

**D**

**Figure 8.**

cells) and macrophages have been identified. In tumour progression, other types of organ education may contribute to B220$^+$CD11c$^+$NK1.1$^+$ cells; further research into this unique phenomenon may reveal additional anti-tumour activity.

# Materials and Methods

### Reagents

The following primary antibodies and factors were used in this study: anti-vitronectin (ab45139, Abcam, Cambridge, MA; MAB38751, R&D Systems, Inc., Minneapolis, MN), anti-thrombospondin 1 (Ab-11, Thermo Scientific, Hudson, NH; ab1823, ab3131, Abcam), anti-FX (H-120 and C-20, Santa Cruz Biotechnology, Santa Cruz, CA), anti-fibrinogen (CSI19761A, Cell Sciences Inc., Canton, MA), anti-albumin (A90-134A, Bethyl Laboratories, Inc., Montgomery, TX), anti-CD45 (ab10558, Abcam; M0701, DAKO, Carpinteria, CA), anti-CD11b (BD Biosciences, San Jose, CA), anti-MECA32 Ab (BD Biosciences), anti-αSMA (M0851, DAKO), anti-CD4 (RM4-5, BD Biosciences), anti-CD8a (53–6.7, BioLegend, San Diego, CA), anti-CD11c (HMα2, BioLegend), anti-B220 (RA3-6B2, BioLegend), anti-NK-1.1 (PK136, BioLegend), anti-TCRβ (H57-597, BD Biosciences) and anti-IFNγ antibodies (Santa Cruz Biotechnology). CCL2, VEGF, TNFα, G-CSF and SDF1 were purchased from R&D Systems (Minneapolis, MN). IL-6 and CXCL1 (Miltenyi Biotec, Bergisch Gladbach, Germany), TGF-β (Peprotech) and HGF Wako Pure Chemical Industries) were used.

### Animals

C57BL6 mice were purchased from Clea Japan (Tokyo, Japan) or SLC (Shizuoka, Japan). FX Friuli$^{-/-}$ mice (Tai *et al*, 2008) and KikGR mice (Tomura *et al*, 2014) were used in experiments. The genetic background of KikGR mouse is C57BL/6. FX Friuli$^{-/-}$ mice were initially introduced as 129Sv/C57BL6/J and backcrossing the offspring with C57BL6/J mice at least three times. To minimize the effect of genetic background, we used littermates in the experiments of FX Friuli$^{-/-}$ mice. Green fluorescent protein (GFP)-Tg (*C57BL/6-Tg (CAG-EGFP)C14-Y01-FM131Osb*) mice were obtained from the RIKEN Bio Resource Center. Mice were housed in a specific pathogen-free condition at a constant temperature and humidity with a 12 h-12 h light–dark cycle. All animal procedures were performed according to the guidelines of the Animal Research Committee of Tokyo Women's Medical University. For animal studies, the investigators were not blinded allocation during experiments and outcome assessment.

### Human samples

All autopsy samples were analysed with approval by the Institutional Review Board of the Tokyo Women's Medical University. The informed consent was obtained from all subjects, and the experiments were confirmed to the principles set out in the WMA Declaration of Helsinki and the Department of Health and Human Services Belmont Report. Autopsy samples of lungs were used for the tumour-free human group ($N = 12$), including patients with dissecting aneurysm, acute cardiac infarction, cerebral infarction, cerebral

haemorrhage and neurodegenerative diseases. Autopsy samples of sets of lungs and livers derived from seven patients with breast cancer were compared with those from twelve tumour-free individuals.

### Photoconversion system for KikGR mice

To detect mobilization of CD45$^+$ cells from the liver to lung, we modified the original method for lymphoid organs (Tomura *et al*, 2008). Briefly, the abdominal skin and peritoneum of anaesthetized KikGR mice were cut to visualize the front lobe of the liver (10 mm in diameter), which was exposed to violet light (436 nm) at 135 mW/cm$^2$ for 4 min using a spot UV-curing equipment (SP500, Ushio, Tokyo, Japan). The exposure was done twice for 2 min while supplying PBS to maintain a wet condition. After closing the abdomen, we examined the liver and lung tissues by flow cytometric and immunohistochemical analyses in TCM-stimulated mice at 24 h, 48 h or 72 h after the photoconversion. Pseudo-surgery was performed on controls. We checked CD11b$^+$ cell mobilization to find out that no significant histological inflammation took place in the liver (data not shown, $N = 4$). We also confirmed that no inflammatory gene expression such as IL-1β, TNF, NFκb1 and NFκb2 occurred in the liver after the exposure (fold change of lung/liver in GSE76506 data). Our histochemical assessment determined that the violet light reached 100–150 μm of depth from the liver surface as shown in a previous study (Kotani *et al*, 2013). We usually used 7- to 9-week male and female mice for TCM injection.

### Tumour cell lines and tumour-conditioned media (TCM)

E0771 breast cancer cells were originally established by Dr. Sirotnak (Memorial Sloan–Kettering Cancer Center, New York, NY) and kindly provided by Dr. Mihich (Roswell Park Memorial Institute, Buffalo, NY) (Ewens *et al*, 2005). Lewis lung carcinoma (LLC) and B16 melanoma cells were purchased from the American Type Culture Collection. Lewis lung carcinoma (3LL) cells were supplied by the Japanese Foundation for Cancer Research (Tokyo). All cells were used during the passages < 6 months and maintained in Dulbecco's modified Eagle's medium (DMEM) supplemented with 10% foetal bovine serum (FBS), 100 units/ml penicillin G sodium and 100 μg/ml streptomycin sulphate. TCM was obtained by incubating cells overnight in serum-free medium.

### Experimental metastatic model

The period of distant primary tumour growth without micro- or macroscopic metastasis was defined to be the pre-metastatic phase. On the other hand, the metastatic phase was defined as the period of tumour cell regrowth in a remote organ, which can be artificially created by an intravenous (i.v.) injection of tumour cells into a tumour-bearing mouse. We used the i.v. injection method to analyse interactions between circulating tumour cells and remote organs. Spontaneous metastasis of LLC, E0771 or B16 cells was microscopically detected in the liver or lungs of tumour-bearing mice only after the primary tumour resection. Thus, for spontaneous lung metastatic assays, we used 3LL cells, a subline of LLC holding highly metastatic ability, or primary tumour resection methods (LLC, E0771 or B16 tumours). Syngeneic tumour grafts were generated via subcutaneous

(s.c.) or mammary fat pad (m.f.p.) implantation of $5 \times 10^6$ tumour cells into 8- to 10-week-old mice. For the tumour cell homing assays, $1–5 \times 10^4$ fluorescent dye (PKH26, Sigma-Aldrich, St. Louis, MO, USA)-labelled metastatic cells were administered to mice, primary tumours of the same size (size-matched), by intravenous injection. Twenty-four–48 h after tumour cell infusion, the lungs were perfused with phosphate-buffered saline (PBS) under physiological pressure to exclude circulating tumour cells. Four to five lung tissue fragments (3 mm in diameter) were randomly excised, and three 10-μm sections per fragment were examined under a confocal (LSM-710, Carl Zeiss MicroImaging GmbH, Germany) or fluorescent microscope (BZ-9000, Keyence, Osaka, Japan). The labelled tumour cell counts were normalized to the total tissue surface area. Age- and sex-matched littermates were used for experiments. We used 6- to 8-week male and female wild-type, $FX^{+/-}$ and $FX^{-/-}$ mice for LLC and E0771 tumours and female for E0771 tumours.

### Bone marrow transplantation

C57BL/6 mice (8-week-old) were lethally irradiated by delivering a 9-Gy fraction to the whole body. The irradiated mice were rescued 24 h later by bone marrow transplants isolated from GFP-Tg mice. Eight weeks after the BMT, we confirmed by flow cytometry that > 95% of the target BMDC had been replaced with GFP-bone marrow cells.

### Isolation of CD45$^+$ cells and B220$^+$CD11c$^+$NK1.1$^+$ cells

To collect tissue containing CD45$^+$ cells, we digested mouse livers and lungs with 0.5 mg/ml of collagenase, 1 mg/ml of dispase and DNase at 37°C for 45 min. Cell suspensions were incubated with mouse CD45-microbeads or NK1.1-microbeads (MACS, Miltenyi Biotec, Auburn, CA), and the captured cells were used for *in vitro* culture or *in vivo* injection. We also isolated photoconverted CD45$^+$ cells obtained from the KikGR-Tg mice using CD45-microbeads and further purified the cells with a fluorescent-activated cell sorter (BD FACSJazz, BD Biosciences, MoFlo Astrios$^{EQ}$, Beckman Coulter, or S3e Cell Sorter, Bio-Rad, Hercules, CA). These experiments using CD45-microbeads are shown in Figs 1A–C and 5D and Appendix Table S2. To obtain B220$^+$CD11c$^+$NK1.1$^+$ cells for culture and *in vivo* injection, we purified the cells with a cell sorter (MoFlo Astrios$^{EQ}$). For flow cytometric analysis, 0.5 μg of Abs per $10^6$ cells in 100 μl volume was used. The percentage in figures is explained in legends. The Zombie Green fixable viability kit (BioLegend) was used to detect dead tumour cells after incubation with B220$^+$CD11c$^+$NK1.1$^+$ cells.

### CD45$^+$ cell culture system using organ tissues

In the organ culture experiments, 2-mm$^2$ tissue specimens were cultured in DMEM without or with FCS (1%) in an upper well culture insert (400-nm pores). CD45$^+$ cells isolated from the organs were cultured in the lower wells, which were stimulated by organ tissues in the upper wells for 24–48 h.

### Immunohistochemistry and cell counts

Anti-Vtn (1:200), anti-TSP (1:300), anti-FX (1:500), anti-fibrinogen (1:200), anti-CD45 (1:500), anti-CD11b (1:200), anti-CD11c (1:100),

anti-CD4 (1:100), anti-CD8 (1:100), anti-NK1.1 (1:100) or anti-B220 (1:100) antibodies were used to stain frozen tissue sections. For negative control staining, we used isotype-matched IgG. The immunostained cell area values are shown as the number of pixels normalized to the DAPI signal. Labelled tumour cells were detected by confocal or fluorescence microscopy and normalized by total surface area.

### Immunoprecipitation and Western blot analysis

Lung tissues were lysed with lysis buffer (50 mM HEPES [pH 7.4], 1% Triton X-100, 150 mM NaCl, 1 mM ethylene glycol tetraacetic acid and 5 mM ethylenediaminetetraacetic acid) supplemented with a protease inhibitor mixture containing 1 mM phenylmethylsulfonyl fluoride, 1 mM sodium fluoride and 1 mM sodium orthovanadate. For immunoprecipitation, the lysate was incubated with anti-fibrinogen Ab, and the resulting complexes were captured with protein G conjugated beads. The beads were washed with PBS + 1% Tween-20, and proteins bound to the beads were subjected to Western blot analysis with anti-fibrinogen (1:2,000), anti-Vtn (1:1,000) or anti-TSP antibodies (1:1,000).

### Microarray analysis

We performed microarray screening using a GeneChip Mouse Genome 230 2.0 Array (Affymetrix). Total RNA was extracted using TRIzol (ThermoFisher) for peripheral blood cells derived from tumour-bearing or tumour-free mice. Complementary RNA, generated from total RNA was hybridized to the Mouse Genome 230 2.0 Arrays following the manufacturer's protocols. Total RNA of photoconverted KikGR cells from liver and lung tissue was isolated using SV Total RNA Isolation System (Promega). Then, cDNA was synthesized using Ovation One-Direct System (NuGEN) and biotinylated cRNA was prepared using the NuGEN Encore Biotin Module (NuGEN) according to the manufacturer's instruction. Next, biotin-labelled cRNA was fragmented and hybridized to the Mouse Genome 430 2.0 Arrays.

Gene expression data were normalized with the MAS5 algorithm using AGCC (Affymetrix® GeneChip® Command Console® Software) and Affymetrix® Expression Console™. A principal component analysis was conducted by inputting the filtered probes that exhibit a score of more than 500 in at least one of six samples.

### Quantitative PCR

Total RNA samples were isolated from frozen tissues and cells using TRIzol reagent (Invitrogen, Carlsbad, CA) and used to synthesize cDNA with reverse transcriptase (SuperScript *VILO*, Invitrogen). cDNA from the sorted cells was amplified using *TaqMan PreAmp* Master Mix (Applied Biosystems, Foster City, CA). Quantitative polymerase chain reaction (PCR) analysis was performed using SYBR Green Master Mix or TaqMan Fast Advanced Master Mix (Applied Biosystems) in a detection system (StepOnePlus, Applied Biosystems). Gene expression levels were calculated from Ct values, and the relationship between the Ct value and the logarithm of the copy number of the target gene was confirmed to be linear using serial dilutions of the corresponding isolated DNA as a standard. Additionally, the gene expression level was normalized to that of

β-actin in each sample. Primer sequences are shown in the Appendix Supplementary Methods. Thus, *TaqMan probes* and *primers* were used to amplify Vtn and Thbs1 and qPCR primers were used to amplify FX.

TaqMan Primer and Probe Sets:

Vtn, NM_011707
Mm00495976_m1
Thbs1, NM_011580
Mm00449032_g1
FX
Mm00484177_m1

q-PCR primers:

F10, NM_007972
5′-AGGTCCGTGAAATCTTCGAG-3′
5′-CATCTCGACACGCTCCTTG-3′

### Cytotoxicity experiments and IFNγ staining

We first set up a detection system of $B220^+CD11c^+NK1.1^+$ cell-dependent cytotoxicity for tumour cells using a flow cytometer. After collecting $B220^+CD11c^+NK1.1^+$ cells that were isolated from liver, the cells were primed with several conditioned media (CMs), which were prepared using TCM-stimulated mouse lungs. Then, rhodamine-labelling tumour cells ($1 \times 10^5$ cells) were cultured with the primed $B220^+CD11c^+NK1.1^+$ cells ($1 \times 10^5$ cells), and thereafter, viable tumour cells were detected using Zombie staining system (Zombie Green fixable viability kit). Alternatively, we counted viable tumour cells using a fluorescent microscope. Intracellular IFNγ expression was stained after perforation of the cell membrane with 0.1% Triton X-100/PBS solution in 2 min.

### Measurement of prothrombin time

We carried out modified method of procedure of HemosIL Recombi-PlasTin (Instrumentation Laboratory) and coagulation assay (Heidtmann & Kontermann, 1998). In briefly, fifty microlitres of normal mouse or tumour-bearing mouse plasma with various dilutions of equal microlitres fibrinogen was prepared to measure prothrombin time (PT). To determine PT, we recorded absorbance at 671 nm after mixing 100 μl 10-fold diluted of HemosIL RecombiPlasTin (as a source of thromboplastin and calcium) with samples. $B220^+CD11c^+NK1.1^+$ cells were washed with Hanks' balanced salt solution (HBSS) prior to use.

### Generation of FX-overexpressing HeELs

For preparation of biotinylated Factor Xa (FX), the solvent of a commercially available FX was exchanged to HEPES buffer (50 mM, pH 7.2), and then, Biotin-sulfo NHS ester was added to the solution, followed with incubation for 60 min at room temperature and gel filtration. $B220^+CD11c^+NK1.1^+$ cells derived from tumour-bearing mouse lung or livers were treated with 20 μM biotin–PEG–lipid for 15 min and then treated with 20 μM neutralized avidin for 5 min

(Kato *et al*, 2004). These cells were treated with biotinylated Factor Xa for 15 min.

### Cell adhesion assay

Fibrinogen (20 mg/ml, Sigma, F4129) was coated on slide culture dishes overnight and washed with PBS three times. To avoid non-specific cell adhesion, boiled aliquots of 0.75% BSA–PBS were used for blocking. Rhodamine-labelled $B220^+CD11c^+NK1.1^+$ cells were primed with various CM and seeded in pre-incubated culture dishes with control rat $IgG_{2a}$ (BD Biosciences), control IgM (MBL), anti-Vtn antibodies (MAB38751, R&D Systems) or anti-TSP antibodies (clone A4.1, IgM, mouse anti-human, Abcam; Amend *et al*, 2015). TSP is highly conserved among species and clone A4.1 has cross-species reactivity with mouse and bovine. Three hours after incubation, wells were washed 5 times with PBS, and attached cells were counted.

### Statistical analysis

Data are expressed as the mean ± SEM. Statistical evaluation was performed as indicated. A $P$-value $< 0.05$ was considered significant. All Welch's *t*-tests were unpaired and two-sided. ANOVA was always used with Bonferroni's correction. We repeated the experiments over two times for representative images. In this study, sample size was not predetermined. Randomization and blinding techniques were not used.

### Study approval

Experiments in mice were carried out with the approval of the Animal Research Committee of Tokyo Women's Medical University. All autopsy samples were analysed with approval by the Institutional Review Board of the Tokyo Women's Medical University.

### Data availability

Microarray data were deposited in NCBI Gene Expression Omnibus as GSE76506 (peripheral blood cells, Appendix Table S1) and GSE76235 (photoconverted KikGR cells, Appendix Table S2). In GSE76235, liver cells designated as liver resident leucocytes (LRLs) are equivalent to HepELs.

**Expanded View** for this article is available online.

### Acknowledgements

This study was supported by Astellas Foundation for Research on Metabolic Disorders (Grant number: 20173312), PRESTO (Precursory Research for Embryonic Science and Technology) "Elucidation and control of the mechanisms underlying chronic inflammation" (S.H.) (Grant number: JPMJPR118B). We thank Drs F. Sirotnak, E. Mihich, J. Miyazaki (Osaka University), A. Miyawaki (RIKEN BSI), M. Murakami, T. Okada (RIKEN IMS), A. Deguchi and K. Ieguchi for providing resources and valuable advice.

### Author contributions

SH and YM designed the experiments. SH, TT, HW, TM, YuM, TO, SI, SY, TH, TY, NS and AW performed the experiments, and SH, TT, SY, HW, MO, AW, HK, HA, MT, KH and YM analysed the data. SH, TT, KH and YM wrote the

**The paper explained**

**Problem**

It has been reported that tumour-infiltrating immune cells including metastasis-associated macrophages (MAMs) exacerbate metastasis. However, anti-metastatic immune cells that affect metastatic niche have been unknown.

**Results**

We used an *in vivo* cell-tracking model in tumour-bearing mouse to discover that B220$^+$CD11c$^+$NK1.1$^+$ cells are entrained in liver to upregulate coagulation factor X (FX) expression and relocated to the lung with anti-tumour activity. These cells eliminate the fibrinogen-rich soil in the pre-metastatic lung to reduce the risk of metastasis. We also found that these cells were able to attack tumour cells directly. Importantly, FX$^+$ leucocytes with a similar expression pattern were detected in clinical samples of lung and liver in the same cancer patients.

**Impact**

The liver-educated B220$^+$CD11c$^+$NK1.1$^+$ cells were found in the pre-metastatic lungs, and an injection of those cells suppressed metastasis in our mouse model.

paper. All authors provided intellectual input and approved the submitted manuscript.

## Conflict of interest

The authors declare that they have no conflict of interest.

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
