## [Review Process File · EMBO Molecular Medicine]

Hepato-entrained B220+CD11c+NK1.1+ cells regulate pre-metastatic niche formation in the lung

Sachie Hiratsuka, Takeshi Tomita, Taishi Mishima, Yuta Matsunaga, Tsutomu Omori, Sachie Ishibashi, Satoshi Yamaguchi, Tsuyoshi Hosogane, Hiroshi Watarai, Miyuki Omori-Miyake, Tomoko Yamamoto, Noriyuki Shibata, Akira Watanabe, Hiroyuki Aburatani, Michio Tomura, Katherine A. High and Yoshiro Maru

Review timeline:

Submission date:	03 November 2017
Editorial Decision:	19 January 2018
Revision received:	20 April 2018
Editorial Decision:	09 May 2018
Revision received:	18 May 2018
Accepted:	23 May 2018

Editor: Céline Carret

Transaction Report:

1st Editorial Decision

19 January 2018

Thank you for the submission of your manuscript to EMBO Molecular Medicine. We have now heard back from the two referees whom we asked to evaluate your manuscript.

The referees find the paper novel and of interest. While referee 1 is rather enthusiastic, referee 2 points to missing mechanism and unexplained / unclear data, leaving the study unconnected. We feel that, should you resolve the issues commented by the referees, this would greatly benefit the paper, improving conclusiveness and clarity.

We would welcome the submission of a revised version within three months for further consideration and would like to encourage you to address all the criticisms raised as suggested to improve conclusiveness and clarity. Please note that EMBO Molecular Medicine strongly supports a single round of revision and that, as acceptance or rejection of the manuscript will depend on another round of review, your responses should be as complete as possible.

I look forward to receiving your revised manuscript.

***** Reviewer's comments *****

Referee #1 (Remarks for Author):

This is a very interesting study on a timely and impactful issue. The authors of this manuscript study the potential role of a "third organ" (in this case the liver) in the control of the premetastatic niche in the lung. These authors show that "liver-educated" leukocytes that they name as HepELs and characterized by B220+CD11c+NK1.1+ surface markers, are critical anti-metastatic cells in the lung. They perform elegant in vitro and in vivo cause-and-effect rigorous experiments that clearly support their hypothesis. In opinion of this reviewer, the paper should be published as it is. The only criticism is that the manuscript would benefit of a better English proofreading that should improve its flow.

Referee #3 (Remarks for Author):

In the manuscript by S. Hiratsuka, the authors have identified a rare NK cells (B220+CD11c+NK1.1+) which are derived in the liver and in turn these cells are recruited to the pre-metastatic lung where these cells support the upregulation of interferon and convert fibronogen to vitronectin that together suppress lung metastasis. This work opens up a new avenue of research supportive of the systemic effects of cancer beyond the crosstalk between the primary tumor, bone marrow and pre-metastatic/metastatic lung. The work emphasizes the need to focus on new explorations to alter pre-metastatic sites preventing metastasis. Despite the extreme novelty of this work there are questions that remain.

It is unclear how tumor cells activate these HepELs. Are the HepELs only found in the liver and where in the liver are they found. It seems the original publication describing these cells describe these NK cells in lymph nodes as well. What does it mean to have Factor X in these specialized NK cells. Are there other coagulation factors packaged inside. Do they have a role in clotting. How are these cells recruited specifically in the lung. Once in the lung, is interferon release derived only from the HepELs or other cells contribute to this process. Interferon can promote the immune system to attack the tumors but also promote inflammation, which can support metastasis. The authors need to explain IFN's role further. How do the HepELs convert fibrinogen to vitronectin. Do these NK cells bind other ECM molecules. What molecular pathways are ongoing here. In most cases of the models used here, metastasis progresses. So when do these cells interfere with metastasis, a time course study (not just 72 hours) on these particular cells in the liver, lung, blood and bone marrow are warranted in these studies. When metastases progresses, do these cells remain or disappear. Do these cells get re-educated, or do other cells override the effects of the HepELs

The knockdown and OE of Factor X in these particular cells will greatly enhance the novelty and function of these cells during metastasis.

In addition, it is very startling that the authors state that B16, LLC and EO771 implanted at primary sites do not promote lung metastasis. It is very common to get lungs mets from B16 and LLC tumors whihc were subcutaneously implanted and for EO771 implanted in the mammary gland to metastasize to the lungs. Furthermore, EO771 cells implanted in the spleen readily grow in the liver.

The authors should comment on the role of HepELs in the primary tumors.

Other comments:

Figure 1B, Lymph nodes should be considered

1E, it appears that FX and CD45 colocalize predominantly, and do not merge that much, comment Figure 2:

2B, it is very strange that the laser technology also photoconverts rare cells within a region, it should be a patch for the entire laser area. Comment

The authors ignore their own data which show that CD4+CD45+ cells also increase in response to TCM and they also have Factor X associated with them. Comment

The authors need to comments on other works on NK cells in particular (such as Andy Moeller and colleagues) that promote the progression of the premetastatic niche.

Figure 3, supplem figure 3, the flow data are not clear and the authors seems to focus on areas where distinct cells are seen. explain the compensation findings here.

Figure 4, no Y axis labels in D

Figure 5, the authors should have isolated these unique cells from the lung itself where they are recruited and not just the liver where they exit with time.

1st Revision - authors' response

20 April 2018

(Next page)

Referee #1 (Remarks for Author):

This is a very interesting study on a timely and impactful issue. The authors of this manuscript study the potential role of a "third organ" (in this case the liver) in the control of the premetastatic niche in the lung. These authors show that "liver-educated" leukocytes that they name as HepELs and characterized by B220+CD11c+NK1.1+ surface markers, are critical anti-metastatic cells in the lung. They perform elegant in vitro and in vivo cause-and-effect rigorous experiments that clearly support their hypothesis. In opinion of this reviewer, the paper should be published as it is. The only criticism is that the manuscript would benefit of a better English proofreading that should improve its flow.

We appreciate for the reviewer's comments. Our manuscript was sent to a proofreading service. Certificate of proofreading issued by the company is attached "For Reviewer data Certification of English editing".

Referee #3 (Remarks for Author):

In the manuscript by S. Hiratsuka, the authors have identified a rare NK cells (B220+CD11c+NKL1+) which are derived in the liver and in turn these cells are recruited to the pre-metastatic lung where these cells support the upregulation of interferon and convert fibronogen to vitronectin that together suppress lung metastasis. This work opens up a new avenue of research supportive of the systemic effects of cancer beyond the crosstalk between the primary tumour, bone marrow and pre-metastatic/metastatic lung. The work emphasizes the need to focus on new explorations to alter pre-metastatic sites preventing metastasis. Despite the extreme novelty of this work there are questions that remain.

Thank you for the reviewer's valuable comments. We responded for the comments as below. The reviewer's original comments, followed by our answers and revised sentences, are shown..

Comments: It is unclear how tumour cells activate these HepELs.

Answer: It has been reported that molecular signals including CCL2, SDF1, IL6, TNF α , VEGF, G-CSF, TGF β , and CXCL1 are released from the primary tumour to function in the remote organs in the pre-metastatic phase (McAllister & Weinberg, 2014, Wang et al., 2017). We tested if any of these factors induces FX expression in HepELs in the tumour-bearing mouse liver. As shown in Fig 7C, CCL2 and CXCL1 strongly induced FX in HepELs.

In the revised manuscript, Fig 7C data and the following sentences are added.

Next we examined whether any molecule derived from primary tumours induces FX expression in HepELs in the pre-metastatic phase. Because it has been reported that the tumour-derived factors such as CCL2, SDF1, IL6, TNF α , VEGF, G-CSF, TGF β , and CXCL1 function in the pre-metastatic phase (McAllister & Weinberg, 2014, Wang et al., 2017), we applied those factors to HepELs in the tumour-bearing mouse liver *in vitro* (Fig. 7C). We found out that CCL2 and CXCL1 strongly induced FX in HepELs. These data indicate involvement of tumour-derived factors in the regulation of HepELs (Fig. 7C).

Comments: Are the HepELs only found in the liver and where in the liver are they found.

Answer: We searched HepELs in various organs in tumour-bearing mice. As shown in Appendix Fig S7, HepELs were prominently found in the lung and liver in no-tumour mouse but in the lung, liver, peripheral blood, and tumour tissue in tumour-bearing mouse. Further qPCR analysis revealed that the cells in the liver in 3mm-tumour-bearing mouse displayed remarkably high levels of FX expression. In 10mm-tumour bearing mouse, HepELs isolated from the lung, liver, and tumour tissue are also showed high expression levels of FX. Our immunohistochemical data showed that liver HepELs in tumour-bearing mouse were observed in a diffusely-scattered pattern.

Comments: It seems the original publication describing these cells describe these NK cells in lymph nodes as well. What does it mean to have Factor X in these specialized NK cells.

Answer: First, CD45 leukocytes in the liver and lung expressed FX and but not CD45 leukocytes in the lymph node. Remarkably, the FX expression levels of CD45 leukocytes in the liver and lung were enhanced in the tumour-bearing state (revised Fig. 1B). Next, B220⁺CD11c⁺NK1.1⁺ cells in various organs such as lung, liver, peripheral blood, bone marrow, lymph node and the primary tumour were investigated. We collected samples 2, 7, and 14 days after the tumour cell implantation; their approximate tumour sizes were 0 mm (2 days), 3 mm (7 days), and 10 mm (14 days) in diameter, respectively. The sorted cells were used for the qPCR analyses (Appendix Fig. S7, upper) We also added the functional analyses data to show FX-dependent coagulation in HepELs as described later.

We added this data shown in revised Appendix Fig. S7 and sentences as below.

B220⁺CD11c⁺NK1.1⁺ cells in various organs such as lung, liver, peripheral blood, bone marrow, lymph node and the primary tumour were investigated. We collected samples 2, 7, and 14 days after the tumour cell implantation; their approximate tumour sizes were 0 mm (2 days), 3 mm (7 days), and 10 mm (14 days) in diameter, respectively. Among them, the FX expression levels in B220⁺CD11c⁺NK1.1⁺ cells isolated from the liver of

3mm tumour-bearing mice were remarkably high (Appendix Fig. S7, upper panel). We would like to note that the FX expressions in the cells derived from the lung and tumour tissues in 10mm tumour-bearing mice were also observed. (Appendix Fig. S7, upper panel).

Comments: Are there other coagulation factors packaged inside. Do they have a role in clotting.

Answer: Based on the array data, coagulation factor 5 and factor 13 as well as factor 10 were packed in peripheral blood leukocytes in tumour-bearing mouse (For reviewer Table). Among them, coagulation factor 10 was upregulated in tumour-bearing mice. To determine the coagulation activity, we calculated prothrombin time (PT) using the B220⁺CD11c⁺NK1.1⁺ cells derived from the liver and lung in tumour-bearing mice. We concluded that B220⁺CD11c⁺NK1.1⁺ cells play a role in clotting because addition of these cells in plasma reduced PT. (data statement was shown in the responses for FX-OE-HepELs).

We added sentences in the revised text as below.

Our microarray data indicates that coagulation factor 5 and factor 13 as well as factor 10 (FX) were packed in peripheral blood leukocytes in tumour-bearing mouse (Appendix Table S1 : GSE76506). Among them, FX was upregulated in tumour-bearing mice. We set up a coagulation assay system to measure prothrombin time (PT) of B220⁺CD11c⁺NK1.1⁺ cells derived from tumour-bearing mice. To determine PT, we recorded absorbance at 671 nm after mixing HemosIL RecombiPlasTin with samples. In our assay, 50 mg/dL of purified fibrinogen showed 10 sec of PT. Then, we examined the effect of B220⁺CD11c⁺NK1.1⁺ cells. The B220⁺CD11c⁺NK1.1⁺ cells (5×10^3 cells) derived from the lung or liver in tumour-bearing mouse showed PTs of 176 sec (lung, n=3) and 190 sec (liver, n=6), respectively. These data imply that the B220⁺CD11c⁺NK1.1⁺ cells play a role in coagulation cascade.

Comment: How are these cells recruited specifically in the lung.

Answer: We consider that HepELs leaving the liver go into the circulatory system, and are trapped in fibrinogen-enriched niche in the lung by an interaction with fibrinogen binding molecules (such as Vtn or TSP, shown in Fig 5) expressed in the cell. A

neutralizing anti-Vtn Ab significantly blocked the binding of liver HepELs to fibrinogen coated plate (Fig 7D). In addition, we confirmed that a neutralizing anti-Vtn and anti-TSP Ab blocked the binding of lung HepELs to a fibrinogen coated plate (Fig 7E). We also tested the binding ability of HepELs to other ECM components as a reviewer requested later (detailed response is shown later)

Comments: Once in the lung, is interferon release derived only from the HepELs or other cells contribute to this process. Interferon can promote the immune system to attack the tumours but also promote inflammation which can support metastasis. The authors need to explain IFN's role further.

Answer: Thank you for your valuable comments. It has been reported that IFN- γ has dual opposite roles as anti-metastatic immune response and promotion of metastatic ability of tumour cells via activated nuclear factor κ B (NF- κ B) signaling pathway (2 ref s). Among immune cells, IFN- γ is produced in CD3⁻NK1.1⁺, CD4⁺ and CD8⁺ T cells and CD3⁺NK1.1⁺ (NKT) cells. In this study, B220⁺CD11c⁺NK1.1⁺ cells showed an anti-metastatic activity by eliminating fibrinogen in a FX-dependent manner in the pre-metastatic phase and by killing tumour cells in the post-metastatic phase. We also have the data showing that CD4⁺T cells, expressing FX and probably producing IFN- γ , promoted metastasis (For reviewer data 1) This result indicates that some activated immune cells might support metastasis.

We added the following sentences in the section of discussion.

It has been reported that IFN- γ has dual opposite roles as anti-metastatic immune response and promotion of metastatic ability of tumour cells via activated nuclear factor κ B (NF- κ B) signaling pathway (Xu, Li et al., 2018, Zhang, Zhu et al., 2011). Among immune cells, it has been reported that IFN- γ is produced in CD3⁻NK1.1⁺, CD4⁺ and CD8⁺ T cells and CD3⁺NK1.1⁺ (NKT) cells. In this study, B220⁺CD11c⁺NK1.1⁺ cells showed an anti-metastatic activity by eliminating fibrinogen in a FX-dependent manner in the pre-metastatic phase and by killing tumour cells in the post-metastatic phase. In addition, we observed that CD4⁺HepELs, expressing FX and probably producing IFN- γ , promoted lung metastasis (unpublished data). Thus, some activated immune cells might support metastasis.

Comments: How do the HepELs convert fibrinogen to vitronectin. Do these NK cells bind other ECM molecules.

Answer: In this study, we found that HepELs express fibrinogen-binding molecules Vtn or Tsp in when they are in the liver or lung, respectively. In addition, it has been reported that accumulation of fibronectin (FN) and crosslinking of collagen I (via lysyl oxidase) provide a platform for the adhesion of BMDCs (Peinado et al., 2017). We examined binding ability of HepELs to other ECM such as collagen I and fibronectin. Our data present that HepELs were able to attach to collagen I or FN although their affinities were not as high as that to fibrinogen (shown in revised Fig.7E). Moreover, the HepEL-Fibronectin/Collagen I interactions were Vtn/TSP independent. We added sentences in the revised text as below.

We focused on TSP as a ligand molecule of fibrinogen because TSP was upregulated in lung HepELs but not in liver HepELs (Appendix Table S2). Our results exhibit that anti-Vtn Ab inhibited binding of lung HepELs to a fibrinogen coated plate. Similarly, neutralizing anti-TSP Ab blocked the binding of lung HepELs (Fig 7E). Then, we examined binding abilities of HepELs to other ECM such as collagen I and fibronectin.(FN) Our data present that HepELs were able to attach to collagen I or FN although their affinities were not as high as that to fibrinogen (Fig.7E). Moreover, the HepEL-FN/collagen I interactions were Vtn/TSP independent.

Comment: What molecular pathways are ongoing here. In most cases of the models used here, metastasis progresses. So when do these cells interfere with metastasis, a time course study (not just 72 hours) on these particular cells in the liver, lung, blood and bone marrow are warranted in these studies. When metastases progresses, do these cells remain or disappear.

Answer: Thank you for the reviewer's comment. As shown above, we found that CCL2 and CXCL1 stimulated FX in HepELs (Revised Fig. 7C). In addition, We investigated involvement of any of transcription factors peculiar to liver by using siRNA transfection or electroporation technique. Our data, shown in "For Reviewer data 2", suggest that FX expression in HepELs are regulated by Foxa1, Cebp- α and Rela, because knockdown of each one of the three transcription factors achieved 35-50% reduction of FX expression. We added a sentence as shown below in discussion in the revised

manuscript.

Deciphering molecular mechanisms responsible for the FX induction in HepELs in a liver environment is ongoing. Based on our data, we speculate that FX expression in HepELs is regulated by multiple transcription factors such as Foxa1, Cebp- α and Rela,(unpublished data).

We analyzed populations of B220⁺CD11c⁺NK1.1⁺ cells in tumour-bearing mouse liver, lung, bone marrow, peripheral blood, lymph node, and tumour tissues to find out that the triple positive cells were observed in tumour-bearing liver, blood and tumour tissues, as stated above. In addition, the sorted cells showed remarkable upregulation of FX in the liver, lung and tumour tissues in the presence of a primary tumour (revised Appendix Fig. S7). However, the number of HepELs gradually decrease during tumour progression (please compare HepELs in 3mm- vs 10mm- size tumour-bearing mice).

Comments: Do these cells get re-educated, or do other cells override the effects of the HepELs

Answer: The HepELs, obtained from primary tumour-stimulated mice and cultured overnight, showed low level of FX expression (please see NoCM column in Fig 7B). We were able to regain the FX expression in the cells by using Liver-CM in the culture media (please see LiCM column). Thus, we consider that the HepELs can be re-educated in terms of FX expression. CD4⁺ cells may have an ability to override the effects of HepELs because they have FX expression in tumour-bearing mice. However, our animal study showed these cells supported metastasis (For reviewer data 1). Thus, it is very difficult to conclude that other cells do override the effects of the HepELs.

Comment: The knockdown and OE of Factor X in these particular cells will greatly enhance the novelty and function of these cells during metastasis.

Answer: The HepELs prepared from FX-knockdown mice (95% knockdown mouse) have very low levels of FX so that the cells can be used as FX knockdown cells. Fig. 8B presents FX knockdown increased in our tumour cell homing assay. Because FX transgenic mouse is not available, we tried to obtain FX-OE equivalent cells in a different way. We succeeded to prepare FX-OE equivalent cells by introducing

recombinant FX conjugated with polyethylene glycol (PEG)-lipid. We tried to obtain FX-overexpressed (OE) HepELs and succeeded to obtain those cells. We first characterized our FX built HepELs in clotting assay to measure prothrombin time (PT). FX-OE HepELs displayed shorter PT in the assay. (please see the revised text shown below). Then, the FX-OE HepELs were applied in the metastasis assay. The number of metastatic rhodamin-labeled tumour cells was decreased in tumour-bearing lungs after injection of lung HepELs and FX-OE HepELs enhanced the inhibitory activity of HepELs (Fig 8C).

We added sentences as below and revised Fig. 8C.

We tried to establish an activated factor X -overexpression system (FX-OE) in B220⁺CD11c⁺NK1.1⁺ cells. Briefly, biotinylated polyethylene glycol (PEG)-lipid and biotinylated recombinant FX were combined with neutralized avidin to tether FX to B220⁺CD11c⁺NK1.1⁺ cells. Then, we characterized the FX-tethered B220⁺CD11c⁺NK1.1⁺ cells in a clotting assay to measure their prothrombin time (PT). The 2.5 x 10³ of FX-OE-B220⁺CD11c⁺NK1.1⁺ cells (FX-OE HepELs) derived from tumour-bearing mouse lungs showed shorter PT than the HepELs without recombinant FX. (56 seconds (n=3) and 193 seconds (n=3), respectively). Next, we compared these two cells in the metastasis assay. The number of metastatic rhodamin-labeled tumour cells was decreased in tumour-bearing lungs after injection of lung HepELs, and FX-OE HepELs enhanced the inhibitory activity of HepELs. (Fig 8C).

Comments: In addition, it is very startling that the authors state that B16, LLC and EO771 implanted at primary sites do not promote lung metastasis. It is very common to get lungs mets from B16 and LLC tumours which were subcutaneously implanted and for EO771 implanted in the mammary gland to metastasize to the lungs. Furthermore, EO771 cells implanted in the spleen readily grow in the liver.

Answer: We would like to thank for the reviewer's attention. We would like to emphasize that in our assay system we have never observed macro- and micro-metastasis in the pre-metastatic organs. Lung metastasis is observed only after the primary tumour resection, when B16, LLC, or EO771 are subcutaneously implanted. Because these cells readily metastasize when they are implanted in the mammary pad or injected into the tail vein, there is no doubt that location of the primary tumour is

one of the most important factors. We also would like to add that 3LL cells, a subline of LLC, relatively easily accomplish lung metastasis even in the case of subcutaneous implantation. We clearly stated this point with references in the revised text and methods section as shown below.

Text

In this study, a key point of our pre-metastatic model system is that spontaneous metastasis from the primary site was observed only after the primary tumour resection, although an intravenous injection of these cells easily attained lung metastasis (Hiratsuka et al., 2002, Hiratsuka et al., 2006, Hiratsuka et al., 2008) (see Methods). In addition, it should be noted that these tumour cells failed to metastasize to the liver.

Methods

The period of distant primary tumour growth without micro- or macroscopic metastasis was defined to be the pre-metastatic phase. On the other hand, the metastatic phase was defined as the period of tumour cell regrowth in a remote organ, which can be artificially created by an intravenous (i.v.) injection of tumour cells into a tumour-bearing mouse. We used the i.v. injection method to analyse interactions between circulating tumour cells and remote organs. Spontaneous metastasis of LLC, E0771 or B16 cells was microscopically detected in the liver or lungs of tumour-bearing mice only after the primary tumour resection. Thus, for spontaneous lung metastatic assays, we used 3LL cells, a subline of LLC holding highly metastatic ability, primary tumour resection methods (LLC, E0771 or B16 tumours).

Comment: The authors should comment on the role of HepELs in the primary tumours.

Answer: In our in vitro study, the HepELs derived from TCM-primed mouse attacked tumour cells in vitro (Fig. 4C-E), suggesting that HepELs potentially have anti-metastatic activity. However, given the fact that the HepELs population in tumor tissue decreased slightly in the primary tumour growth (Appendix Fig. S7), the effect of HepELs is limited.

We added sentences as shown below in discussion of the revised text.

The B220⁺CD11c⁺NK1.1⁺HepELs derived from TCM-primed mouse attacked tumour

cells in vitro (Fig. 4C-E). Moreover, they were found in the primary tumor, and they became FX⁺ in the later stage of tumour (Appendix Fig. S7). These data suggest that HepELs potentially have anti-metastatic activity. However, given the fact that the HepELs population in tumor tissue decreased slightly in the primary tumour growth (Appendix Fig. S7), the effect of HepELs against the primary tumour is limited.

Other comments:

Figure 1B,

Comment: Lymph nodes should be considered

Answer: Thank you for your suggestion. We added this data in Fig. 1B and revised Appendix Fig. S7

Comment: 1E, it appears that FX and CD45 colocalize predominantly, and do not merge that much,

Answer: FX expression was observed both intra- and extra-cellular region in CD45+ cells. FX was immunohistochemically detected merged area in small-size tumour-bearing mouse lung (Reviewer data 3, upper). During primary tumour progression, abundant FX expression was found in CD45+ cells that may become partially merged image with FX and CD45 (Reviewer data 3, lower). Based on these data, abundant secreted FX may exceed surface of leukocytes stained with CD45 antibody in fibrinogen-rich area in large-size tumour-bearing lungs as shown in Fig. 1E.

Figure 2:

Comment: 2B, it is very strange that the laser technology also photoconverts rare cells within a region, it should be a patch for the entire laser area.

Answer: We used deep UV lamp with a 436 nm bandpass filter as light source. Resulting blue light was transmitted by an optical fiber which allows us to irradiate a particular region (a circle of 10-12 mm in diameter and 100-150 μ m in depth) in the liver. Figure 2B data were taken deep inside the liver so that the photoconversion did not occur in the liver stromal cells. We have observed the photoconversion in the surface liver cells (data not presented in this paper).

Comment: The authors ignore their own data which show that CD4+CD45+ cells also

increase in response to TCM and they also have Factor X associated with them.

Answer: Thank you for your comment. Our data show that CD4⁺T cells, which expressed FX, promoted metastasis (For reviewer data1). These results indicate that some activated immune cells might support metastasis. We added the statement in the discussion of revised manuscript.

Comment: The authors need to comments on other works on NK cells in particular (such as Andy Moeller and colleagues) that promote the progression of the premetastatic niche.

Answer: Thank you for your comment. We added the following sentence in the revised manuscript.

In addition, NK cell-type cytotoxic capacities of CD3⁻NK1.1⁺ cells was reduced by hypoxic primary tumour-derived factors in the pre-metastatic niche (Sceneay et al., 2012)..

Comment: Figure 3, suplem figure 3, the flow data are not clear and the authors seems to focus on areas where distinct cells are seen. explain the compensation findings here.

Answer: KikGR is a powerful tool which allows us to label cells of interest, but it takes 16 min to attain full green-to-red conversion in cells. Short time light exposure produced partial conversions, so that in such a case cells gave green and red double positive signals in the dot plot when they were subjected to a flowcytometric analysis.

For your convenience, data in the reference paper (Cytometry Part A, 87A, 830-842, 2015) are presented in “For reviewer data 4”. Since the data were plotted after the unmixing procdedure, the figure does not necessarily show accurate dot plot pattern. Nevertheless, we consider that this figure well reproduce the behavior of photoconverted cells in the dot plot. This figure clearly displays that short exposure (1-15s) results in upward shift of the photoconverted cells in the dot plot.

In this study, all the photo-converted cells are expected to receive light exposure less than 5 min, because they are moving in the liver during the light exposure. This indicates that the photo-converted cells hold mixture of KikGR-green and kikGR-red.

Given the estimated light exposure time for each cell, vast majority was small amount of kikGR-red and large amount of kikGR-green. The cells locate slightly more upward than pure KikGR-green cells in the dot plot. In order to detect the cells holding small amount of kikGR-red, we set a region shown in Fig 3B. We also would like to note that we empirically confirmed that cells appeared in the region only after the light exposure.

Comment: Figure 4, no Y axis labels in D

Answer: Thank you for your comment. We added label in Fig. 4D

Comment: Figure 5, the authors should have isolated these unique cells from the lung itself where they are recruited and not just the liver where they exit with time.

Answer: Thank you for the reviewer's comment. We added data that B220⁺CD11c⁺NK1.1⁺ cells from tumour-bearing mouse lung (lung HepELs) attached to fibrinogen via TSP (revised Fig. 7E), and suppressed metastatic tumour cell homing in the lungs (revised Fig. 8C).

For reviewer Table

Gene Title	Gene Symbol	RefSeq	Normal_Signal	B16_Signal	LLC_Signal
		Transcript ID			
fibrinogen, alpha polypeptide	Fga(F1)	NM_010196	9.8	1.5	1.9
fibrinogen, B beta polypeptide	Fgb(F1)	NM_181849	1.6	1.6	10.4
fibrinogen, gamma polypeptide	Fgg(F1)	NM_133862	2.2	1.4	3.7
coagulation factor II	F2	NM_010168	5.6	27.9	24.6
coagulation factor III	F3	NM_010171	1.7	0.8	0.5
coagulation factor IX	F9	NM_007979	7.2	12	8
coagulation factor V	F5	NM_007976	2764.6	1922.2	2284.6
coagulation factor VII	F7	NM_010172	17.2	40.8	98.9
coagulation factor VIII	F8	NM_007977	11.7	15.3	2.6
coagulation factor X	F10	NM_007972	252.4	404.6	435.9
coagulation factor X	F10	NM_007972	73.7	121.8	191
coagulation factor X	F10	NM_007972	4.8	54.1	28.6
coagulation factor XI	F11	NM_028066	18.5	11.2	18.3
coagulation factor XII (Hageman factor)	F12	NM_021489	2.7	3	2.6
coagulation factor XIII, A1 subunit	F13a1	NM_028784	3049	3765.1	5071.7
coagulation factor XIII, beta subunit	F13b	NM_031164	1.6	0.7	6.5

For reviewer data 1, (unpublished data in discussion)

1, Tumor-bearing mice with tumor-bearing CD4+ HepELs

2, Tumor-bearing mice

Metastatic tumour cell homing with tumor-bearing CD4+ HepELs in lungs

Rhodamine-labeled E0771 cell homing in the lungs after injection of CD4+ HepELs (1) or no HepELs (2) in E0771-bearing mice. N=6

For Reviewer data 2, (unpublished data in discussion)

Accell									nucleofector			
1	2	3	4	5	6	7	8	9	10	11	12	13
Non-target SiRNA (Accell)	Hnf1a	Hnf4a	Foxa1	Foxa2	CEBP- α	Rela	Sp1	Ap2a	Non-target SiRNA (nucleofector)	Foxa3	GATA6	CEBP- δ

FX expression in liver CD45 cells treated with siRNAs.

One of the two different siRNA delivery systems was used for each siRNA (transfection “Accell” and electroporation “nucleofector”) and compared to non-target siRNA. Numbers showing various siRNAs in the lower part are indicated in the upper graphs. The number in the left column of each graph shows relative mRNA levels normalized by β -actin. N = 3

For reviewer data 3

FX expression was observed both intra- and extra-cellular region in CD45, surface marker of leukocytes, -positive cells. FX was immunohistochemically detected merged area in small-sized tumour-bearing mouse lung (upper). During primary tumour progression, abundant FX expression was found in CD45+ cells that may become partially merged image with FX and CD45 (lower).

For reviewer data 4

Detection of spectral changes of Kaede and KikGR expressing cells during photoconversion. Cells were analysed after violet light irradiation for 0, 1, 5, 15, 30, 70, 120, 240, or 960 sec. KikGR-Green and KikGR-Red intensities after Spectral Unmixing are merged and shown in a single dot-plot. This figure is taken from the reference (Cytometry Part A, 87A, 830-842, 2015).

For reviewer data Certification of English editing

CERTIFICATE OF ENGLISH EDITING

This document certifies that the paper listed below has been edited to ensure that the language is clear and free of errors. The edit was performed by professional editors at Editage, a division of Cactus Communications. The intent of the author's message was not altered in any way during the editing process. The quality of the edit has been guaranteed, with the assumption that our suggested changes have been accepted and have not been further altered without the knowledge of our editors.

TITLE OF THE PAPER

Hepato-entrained B220+CD11c+NK1.1+ cells regulate pre-metastatic niche formation in the lung

AUTHORS

Sachie Hiratsuka

JOB CODE

DLPED_1

Signature

Vikas Narang,
Vice President, Author Services, Editage

Date of Issue
April 13, 2018

Editage, a brand of Cactus Communications, offers professional English language editing and publication support services to authors engaged in over 500 areas of research. Through its community of experienced editors, which includes doctors, engineers, published scientists, and researchers with peer review experience, Editage has successfully helped authors get published in internationally reputed journals. Authors who work with Editage are guaranteed excellent language quality and timely delivery.

CACTUS

Contact Editage

Worldwide

request@editage.com
+1 877-334-8243
www.editage.com

Japan

submissions@editage.com
+81 03-6868-3348
www.editage.jp

Korea

submit-
korea@editage.com
1544-9241
www.editage.co.kr

China

fabiao@editage.cn
400-005-6055
www.editage.cn

Brazil

contato@editage.com
0800-892-20-97
www.editage.com.br

Taiwan

submitjobs@editage.com
02 2657 0306
www.editage.com.tw

2nd Editorial Decision

09 May 2018

Thank you for the submission of your revised manuscript to EMBO Molecular Medicine. We have now received the enclosed reports from the referees that were asked to re-assess it. As you will see the reviewers are now globally supportive and I am pleased to inform you that we will be able to accept your manuscript pending final editorial amendments.

Please submit your revised manuscript within two weeks. I look forward to seeing a revised form of your manuscript as soon as possible.

I look forward to reading a new revised version of your manuscript as soon as possible.

***** Reviewer's comments *****

Referee #3 (Remarks for Author):

The revised version of the manuscript has improved greatly.
The authors have addressed all the queries of this reviewer

2nd Revision - authors' response

18 May 2018

Authors made requested editorial changes.

Corresponding Author Name: Sachie Hiratsuka, Yoshiro Maru

Manuscript Number: EMM-2017-08643